# A Theory of Earthquake Prediction

Jeen-Hwa Wang
Institute of Earth Sciences, Academia Sinica, Taipei, Taiwan, ROC
E-mail: jhwang@earth.sinica.edu.tw

## Abstract

In this study, the pre-seismic strain of an earthquake is considered as a fundamental and important precursor. Based on the Voight's equation for material failure, we theoretically investigate the physical basis on predicting the failure time, magnitude, and location of a forthcoming earthquake in terms of pre-seismic strains generated on or near the related fault where the event will happen. The log(*T*)–*M* relationship is built up. Results exhibit that the failure time depends on the strain rate and two parameters of the Voight's equation; while the magnitude is associated with the precursor time, two parameters of the Voight's equation, and the exponent of the scaling law between the strain and the fault length. The location of the forthcoming earthquake may be qualitatively estimated from the localities of observation sites where the pre-seismic strains are observed. In addition, the anomalous geoelectric and geochemical signals prior to earthquakes are also taken into account as precursors. Their log(*T*)–*M* relationships are derived. The precursor times of geoelectric signals and those of the geochemical signals are, respectively, the same and shorter than that of the pre-seismic strains.

**Keywords**: Earthquake prediction, strain, failure time, magnitude, location, Voight's equation, fault length

# 1 Introduction

The ruptures of earthquakes, especially for large ones, are usually preceded by complex physical and chemical processes which may produce the so-called precursors (e.g., Atkinson, 1984; Main and Meredith, 1989; Main, 1999; Zaccagnino and Doglioni, 2022). Hence, a significant way to reduce seismic hazards is the prediction of forthcoming earthquakes based on observations of reliable precursors. Since Milne (1880) first addressed this viewpoint in the nineteenth century, earthquake prediction has been a challenging problem for earthquake scientists (e.g., Knopoff, 1996). Aki (1989, 2009) assumed that earthquakes are predictable and earthquake scientists should inform the probability of the occurrence of an earthquake with a specified magnitude, place, and time window to the government and the public for mitigating hazards. Although the earthquake prediction seems successful for few large events, including the 1975 Haicheng, China, earthquake (cf. Wang et al., 2006), it has been long a debatable problem of earthquake science. Numerous earthquake scientists address that earthquakes can be predicted, but some others stand for the opposite viewpoint (e.g. Geller, 1997; Geller et al., 1997). The latters were mainly based on the reasons that the brittle crust is quite disordered and complicated (cf. Savage et al., 2010) and it sometimes exists in the critical state (cf. Bak, 1996). The two conditions will reduce the predictability of forthcoming earthquakes. However, disorder and complexity within a single fault could be much lower than those in the brittle crust or a fault system. A fault could be at the subcritical state (cf. Atkinson, 1984; Main and Meredith, 1989) before its failure occurs. Hence, it is still significant to explore an acceptable, workable model for predicting the failure time, $t_f$, the magnitude, $M$, and the source area of a forthcoming earthquake from observed precursors, especially for a single fault.

Although reliable precursors may provide us a clue to judge whether or not an earthquake will happen in an area, the observations of precursors that are merely on the reduction side of science (see Kuhn, 1962) thus cannot be directly applied to predict anything. Hence, earthquake scientists need workable theories or models, which are on the deduction side, for prediction. Up to date, the reduction side is much stronger than the deduction one on the earthquake prediction research. This cannot make earthquake prediction be successful. A major effort is still needed in the

scientific community in order to advance physical theories and models towards the great goal of earthquake prediction. One of the most important matters is the construction of physico-chemical models for respective precursors or even a unified model for all precursors. Through the comparison between the observations and the models, earthquake scientists could obtain the optimum ones for respective precursors or the optimum unified one. Based on the optimum models or the optimum unified one, earthquake scientists may be capable of predicting an earthquake, including its location, time window, and magnitude as mentioned above. Of course, such a model could be region-dependent, because different tectonic and geological conditions will influence the parameters of the model.

Reid's elastic rebound theory (Reid, 1910) assumes that the loading stress and slip on a fault are the major factors in causing an earthquake rupture. Numerous authors (e.g., Dieterich, 1978; Lomnitz and Lomnitz-Adler 1981; Kostrov and Das, 1982; Main, 1988, 1999; Scholz, 1990) assumed that the pre-seismic stress, $\sigma$, and slip, $u$ (or strain, $\varepsilon$), on a fault are two important factors in influencing the generation of precursors. Anomalous pre-seismic displacements or strains near the faults have been observed before numerous earthquakes. Tsubokawa et al. (1964) first measured pre-seismic displacements at several inland sites before the June 16 1964 M7.5 Niigata, Japan, earthquake. Kanamori (1973, 1996) reported pre-seismic release associated with forthcoming major earthquakes, especially in Japan. Yu et al. (2001) reported the pre-seismic displacements on the near-fault stations before the September 20 1999 $M$7.6 Chi-Chi, Taiwan, earthquake. Papazachos et al. (2002) found accelerating pre-seismic crustal deformation before large earthquakes in the Southern Aegean area. Sarkar (2011) observed possible accelerated Benioff strains prior to large earthquakes in the Sistan Suture Zone of Eastern Iran. These studies confirm the significance and importance of pre-seismic slip or strain on either earthquake prediction or assessment for forthcoming earthquakes. These studies confirm the significance and importance of pre-seismic slip or strain on earthquake prediction or assessment of forthcoming earthquakes.

Laboratory experiments reveal that $\sigma$ and $u$ are time-varying (Atkinson, 1984; Rudnicki, 1988; Main and Meredith, 1989). While, the slip as well as the strain increased very slowly with time from the initial time $t_0$ to a particular time $t_c$ and then increased rapidly from $t_c$ up to the failure time $t_f$ when an earthquake happens This is

the so-called quasi-static subcritical crack growth (SCG) model (Atkinson, 1984, 1987; Atkinson and Meredith, 1987) which is usually represented by the Charles law (e.g., Das and Scholtz, 1981; Main, 1988, 1999). Das and Scholz (1981) used this model with Charles law to describe the acceleration of a crack tip from an initially slow (sub-critical) rate due to stress corrosion to rapid remarkable rupture under increasing stresses. They predicted the failure time which depends on initial conditions on a fault, such as crack length, crack-tip velocity, residual frictional stress following a previous earthquake, stress-corrosion index, and the rate of stress input. Main (1988) applied a similar theory to predict the occurrence time of an event. His model may quantitatively explain the decrease of failure time in the crust in terms of decreases in the residual stress due to increasing heat flow, coupled with increases in both stress-input rates and density of nucleation points for rupture initiation. The model also predicts progressively increasing failure times for normal, strike-slip, and thrust faults under similar conditions. Wang (2021a,b; 2023) and Wang et al. (2016) classified the long-term, intermediate-term, short-term, and immediate-term precursors based on the SCG (subcritical crack growth) model as mentioned above.

From rock mechanic experiments, Voight (1988, 1989) proposed a nonlinear rate-dependent law for material failure:

$$X_{tt}\text{-}aX_t^{-\alpha}=0 \quad (1)$$

where $X$ is an observable quantity, $X_{tt}$ *and* $X_t$ denote $d^2X/dt^2$ and $dX/dt$, respectively, $a$ is a constant, and $\alpha$ is the scaling exponent of the model. Based on rock mechanics, $X$ may be interpreted in terms of conventional geodetic observations (e.g., length change, fault slip, strain or angular change), seismic quantities (e.g., the square root of cumulative energy release or Benioff strain) or geochemical observations (such as gas emission rates or chemical ratios). The parameter $\alpha$ varies with rock materials and also depends on the temperature. Eq. (1) is called the Voight's equation hereafter. Some authors (e.g., Varnes, 189; Kilburn and Voight, 1998) compared Eq. (1) with the Charles law for the SCG model. Essentially, the Voight's equation is similar to the Charles law. The Voight's equation has been applied to predict the failure time of an earthquake based on the accelerated Benioff strain (e.g., Bufe and Vanus, 1993; Bowman et al., 1996) and the accelerating strain (e.g., Main, 1999). In addition, Main

(1999) also studied the failure times of earthquakes by considering constitutive rules of a simple percolation model (e.g., Stauffer and Aharony, 1994). However, they did not predict the magnitude of a forthcoming earthquake.

The pre-seismic strains observed on or near a fault are directly related to the stress and slip on the fault zone. Define $T=t_f-t_0$, where $t_0$ is the initial occurrence time of the precursor, be the precursor time (see Wang et al., 2016; Wang, 2021a,b). In this study, we will propose a theory to predict the failure time, $t_f$, magnitude, $M$, and location of a forthcoming earthquake and to investigate the relationship between the precursor time and earthquake magnitude from the pre-seismic and co-seismic strains based on the Voight's equation. In addition, the theory can be also applied to other kinds of precursors.

## 2 Voight's Equation

From the results obtained from the rock mechanics experiments, Voight (1988) proposed the empirical equation, i.e., the so-called Voight's equation, to describe rate-dependent material failure. The Voight's equation has been considered as a fundamental physical law governing diverse forms of material failures (e.g. Voight, 1988, 1989). It is a more general form of Charles' law (Main, 1999). Like several authors (e.g., Das and Scholtz, 1981; Main, 1988, 1999), I assume that this empirical equation can be applied to real earthquakes. In addition, this empirical equation has been applied to volcanic eruptions (Voight, 1988b; Cornelius and Voight, 1995; Kilburn and Voight, 1998).

If $X$ in Eq. (1) is taken to be the strain, $\varepsilon$, on a fault, the final stages of failure under steady conditions of a rock in compression would show a proportionality between the logarithm of creep acceleration and the logarithm of creep velocity. Integrating Eq. (1) gives the expression for the strain rate, $\varepsilon_t$, and strain acceleration, $\varepsilon_{tt}$, on a fault zone. In the followings, the strain and strain rate at the initial time, $t_o$, are denoted by $\varepsilon_o$ and $\varepsilon_{to}$, respectively; while those at the failure time, $t_f$, are shown by $\varepsilon_f$ and $\varepsilon_{tf}$, respectively. The solution is dependent on the scaling exponent $\alpha$. For $\alpha$=1, the strain rate is

$$\varepsilon_t=\varepsilon_{to}e^{a(t-to)}. \quad (2)$$

For $\alpha<1$, the strain rate is

$$\varepsilon_t=[a(1-\alpha)(t-t_o)+\varepsilon_{to}{}^{(1-\alpha)}]^{1/(1-\alpha)}. \tag{3}$$

For $a>1$, the strain rate is

$$\varepsilon_t=[a(\alpha-1)](t_f-t)+\varepsilon_{tf}{}^{(1-\alpha)}]^{1/(1-\alpha)}. \tag{4}$$

These equations remarkably reveal that $\varepsilon_t$ increases with time and thus there is not an upper bound of $\varepsilon_t$. The value of $\varepsilon_t$ can be evaluated from the first two equations for $\alpha\leq1$ and cannot be resolved from the third equation for $\alpha>1$. It seems that there is a singular point at $t_f$ for $\alpha>1$. At the singular point, a rock fracture or an earthquake would happen. An example of numerical results can be seen in Voight's (1989) Figure 2. Since $\varepsilon$ is integrated from $\varepsilon_t$, there is not an upper bound value for $\varepsilon$ when $\alpha\leq1$.

We may further solve the time-dependent strain $\varepsilon(t)$ through double integration of Eq. (1). For $\alpha>1$ and $\alpha\neq2$, the result is

$$\varepsilon(t)-\varepsilon_o=\{[a(\alpha-1)(t_f-t_o)+\varepsilon_{tf}{}^{(1-\alpha)}]^{\eta}-[a(\alpha-1)(t_f-t)+\varepsilon_{tf}{}^{(1-\alpha)}]^{\eta}\}/a(\alpha-2) \tag{5}$$

where $\eta$ represents $(2-\alpha)/(1-\alpha)$. For $\alpha>1$ and $\alpha\neq2$, the values of $\eta$ are: (1) $\eta<0$ as $1<\alpha<2$; and (2) $\eta>0$ as $\alpha>2$. From the theoretical studies made by Main (1998), we can see that the condition of the existence of accelerating strain for generating an earthquake is $1<\alpha<2$, thus leading to $\eta<0$. This condition will be used hereafter.

## 3. Theory of earthquake prediction

According to the Voight's equation, I assume that it is possible to predict the failure time of a forthcoming earthquake from the observed pre-seismic strains measured on or near the fault along which the event will occur. The prediction of the failure time is based on Eq. (4) and the prediction of the magnitude is based on Eq. (5). The location of the event should be near the sites of observing the pre-seismic strains. The theory

of earthquake prediction proposed in this study is described below.

### 3.1 Predicting the Failure Time of a Forthcoming Earthquake

Since the condition $1<\alpha<2$ is considered here, we will only take Eq. (4) in the followings. Due to $1-\alpha<0$, the strain rate, $\varepsilon_{tf}$, at the failure time should be much larger than 1 strain/sec and thus $\varepsilon_{tf}^{1-\alpha}$ is much smaller than 1 strain/sec. This makes Eq. (4) become

$$\varepsilon_t=[a(\alpha-1)(t_f-t)]^{1/(1-\alpha)}. \tag{6}$$

The time variations in $\varepsilon_t$ from Eq. (6) for $\alpha$=1.5, 1.6, and 1.7 when $a$=0.5 are displayed in Fig. 1 in which $\varepsilon_t$ is normalized by the maximum value of $\varepsilon_t$ for the three cases. In the figure, the three curves intersect to one another at a point with $t=t_c$. When $t<t$, $\varepsilon_t$ increases slowly with time and increases with $\alpha$; while when $t>t$, $\varepsilon_t$ increases rapidly with time and decreases with increasing $\alpha$.

From Eq. (6), we propose a method to explore the possibility of predicting the failure time, $t_f$ of a forthcoming earthquake. Since the values of three model parameters $t_f$, $a$, and $\alpha$, must be solved, those of $\varepsilon_t$ at three time instants should be given. Considering the pre-seismic strain rates, i.e., $\varepsilon_{t1}$, $\varepsilon_{t2}$, and $\varepsilon_{t3}$, at three time instants, i.e., $t_1$, $t_2$, and $t_3$, respectively. An example for $\alpha$=1.6 with $a$=0.5 is shown in Fig. 2 in which $\varepsilon_t$ is normalized by the maximum value of $\varepsilon_t$. Inserting $\varepsilon_{tj}$ and $t_j$ ($j$=1, 2, and 3) into Eq. (6) yields

$$\varepsilon_{tj}=[a(\alpha-1)(t_f-t_j)]^{1/(1-\alpha)} \quad (j=1, 2, 3). \tag{7}$$

This leads to

$$t_f=t_j+\varepsilon_{tj}^{(1-\alpha)}/a(\alpha-1) \quad (j=1, 2, 3). \tag{8}$$

From Eq. (8) for $\varepsilon_{t1}$ at $t_1$ and $\varepsilon_{t2}$ at $t_2$, we have

$$t_2\text{-}t_1=[\varepsilon_{t2}^{(1\text{-}\alpha)}\text{-}\varepsilon_{t1}^{(1\text{-}\alpha)}]/a(\alpha\text{-}1) \quad (9a)$$

or

$$a(\alpha\text{-}1)=[\varepsilon_{t2}^{(1\text{-}\alpha)}\text{-}\varepsilon_{t1}^{(1\text{-}\alpha)}]/(t_2\text{-}t_1). \quad (9b)$$

Similarly, from Eq. (8) for $\varepsilon_{t1}$ at $t_1$ and $\varepsilon_{t3}$ at $t_3$ we have

$$t_3\text{-}t_1=[\varepsilon_{t3}^{1/(1\text{-}\alpha)}\text{-}\varepsilon_{t1}^{1/(1\text{-}\alpha)}]/a(\alpha\text{-}1) \quad (10a)$$

or

$$a(\alpha\text{-}1)=[\varepsilon_{t3}^{(1\text{-}\alpha)}\text{-}\varepsilon_{t1}^{(1\text{-}\alpha)}]/(t_3\text{-}t_1). \quad (10b)$$

Define two functions in term of $\alpha$, i.e., $F_{21}(\alpha)=[\varepsilon_{t2}^{(1\text{-}\alpha)}\text{-}\varepsilon_{t1}^{(1\text{-}\alpha)}]/(t_2\text{-}t_1)$ and $F_{31}(\alpha)=[\varepsilon_{t3}^{(1\text{-}\alpha)}\text{-}\varepsilon_{t1}^{(1\text{-}\alpha)}]/(t_3\text{-}t_1)$. From Eqs. (9b) and (10b), $F_{21}(\alpha)$ and $F_{21}(\alpha)$ are the same because they are both equal to $a(1\text{-}\alpha)$. We may evaluate the value of $\alpha$ directly from the equality $F_{21}(\alpha)=F_{31}(\alpha)$. We first plot the difference of the two functions for $1<\alpha<2$. An example of $F_{21}(\alpha)\text{-}F_{21}(\alpha)$ in terms of $\alpha=1.6$ is shown in Fig. 3 in which the normalized values of $F_{21}(\alpha)\text{-}F_{21}(\alpha)$, i.e., $(F_{21}(\alpha)\text{-}F_{21}(\alpha))/(F_{21}(\alpha)\text{-}F_{21}(\alpha))_{\max}$, is given. The condition for the existence of the value of $\alpha$ to make $F_{21}(\alpha)=F_{21}(\alpha)$ is that the curve of $F_{21}(\alpha)\text{-}F_{31}(\alpha)$ must intersect the horizontal line with $F_{21}(\alpha)\text{-}F_{31}(\alpha)=0$ at a point with a certain value of $\alpha$ as displayed in Fig. 3. After the value of $\alpha$ has been evaluated, we may calculate the value of $a$ from either $a=F_{21}(\alpha)/(1\text{-}\alpha)$ or $a=F_{31}(\alpha)/(1\text{-}\alpha)$. Then, we may evaluate the failure time of the forthcoming earthquake from Eq. (7) by using the following expression:

$$t_f=t_j+\varepsilon_{tj}^{1/(1\text{-}\alpha)}/a(\alpha\text{-}1) \quad (j=1, 2, 3). \quad (11)$$

The difference between the occurrence time of a precursor and the failure time of the forthcoming earthquake is called the precursor time (e.g., Wang et al., 2016; Wang, 2021a,b) and is denoted by $T$ hereafter. For the present case, the occurrence time of

the precursor and the failure time of the forthcoming earthquake are $t_o$ and $t_f$, respectively, thus leading to $T=t_f-t_o$.

### 3.2 Prediction of the Magnitude of a Forthcoming Earthquake

Based on the evaluated precursor time, $T$, it is possible to predict the magnitude of an earthquake by using Eq. (5). It first needs to discuss the value of initial strain $\varepsilon_o$. After the ruptures of last earthquake on a fault, the fault usually continues to slide with the relative movement speed of regional plates until the occurrence of the next event. If the moving speed is $v_p$, the strain rate, $\varepsilon_t$, is $v_p/L$ where $L$ is the fault length on a fault. Here the value of $\varepsilon_t\delta t$ with the time unit $\delta t$ of 1 second is taken to be $\varepsilon_o$. The value of $\varepsilon_t$ is commonly $10^{-6}$ strain/year around the world (e.g., Scholz et al., 1973; Turcotte and Schubert, 1982; Yu et al., 2001). For an example, the value of $\varepsilon_t$ is $0.25\times10^{-6}$/yr=$1.90\times10^{-13}$/sec for the San Andres fault (cf. Turcotte and Schubert, 1982), thus leading to $\varepsilon_o=1.90\times10^{-13}$ which is much smaller than 1. This makes us able to take $\varepsilon_o=0$ in this study. Figures 1 and 2 reveal $\varepsilon_{tf}>>1$. According to the two conditions, Eq. (5) becomes

$$\varepsilon(t)=\{[a(\alpha-1)T]^{\eta}-[a(\alpha-1)(t_f-t)]^{\eta}\}/a(\alpha-2). \qquad (12)$$

Voight (1988, 1989) took $a$=0.5 for studying the results of his rock mechanic experiments. Hence, the values of $\varepsilon_t$ is about a few strain/day or $10^{-4}$ strain/sec for laboratory earthquakes. As mentioned above, the values of pre-seismic strain, $\varepsilon$, much before the occurrences of natural earthquakes are very small. Hence, the value of parameter $a$ should be small for natural earthquakes. Nevertheless, the value of $a$ is still taken to be 0.5 in Fig. 4 which illustrates the time variations in $\varepsilon$ from Eq. (12) for $\alpha$=1.5, 1.6, and 1.7. In the figure, $\varepsilon$ is normalized by the maximum value of the three cases. Like Fig. 1, Fig. 4 shows that the three curves intersect to one another at a point with $t=t_c$. When $t<t$, $\varepsilon_t$ increases slowly with time and increases with $\alpha$; while when $t>t$, $\varepsilon_t$ increases rapidly with time and decreases with increasing $\alpha$.

The earthquake ruptures at $t=t_f$ when the strain is $\varepsilon_f$, which is

$$\varepsilon_f = [a(\alpha\text{-l})T]^{\eta}/a(\alpha\text{-}2) \tag{13}$$

from Eq. (12). This is the upper bound of $\varepsilon(t)$ for $\alpha>1$ and $\alpha\neq2$. This upper bound is dependent on both the parameters of fault-zone materials and precursor time. Re-organizing Eq. (13) and taking the logarithm on the two sides of the re-organized equation yield

$$\log(T)=\log\{[a(\alpha\text{-}2)\varepsilon_f]^{1/\eta}/[a(\alpha\text{-l})]^{\eta}\}. \tag{14}$$

Eq. (14) gives

$$\log(T)=\log\{[a(\alpha\text{-}2)]^{1/\eta}/a(\alpha\text{-l})\}+\log(\varepsilon_f)/\eta. \tag{15}$$

This represents the power-law scaling relationship between $T$ and $\varepsilon_f$, i.e., $T\sim\varepsilon_f^{1/\eta}$.

Since the rupture duration of an earthquake is short, we may consider $\varepsilon_f$ as the average strain over the ruptured area after failure. Shaw (2023) inferred the scaling law for $\varepsilon_f$ versus the fault length of an earthquake, $L$, in the following form: $\varepsilon_f=\lambda L^{-1/2}$. This leads to

$$\log(\varepsilon_f)=\log(\lambda)\text{-}\log(L)/2, \tag{16}$$

where $\lambda$ is a region-dependent constant. Several authors (e.g., Kanamori and Anderson, 1976; Wells and Coppersmith, 1994; Leonard, 2010; Thingbaijam et al., 2017; Wang, 2018; Shaw, 2023) inferred the scaling law for $L$ versus $M$, which is the earthquake magnitude (usually the seismic-wave magnitude, $M_s$, or the moment magnitude, $M_w$), in the following form:

$$\log(L)=\chi+M/2 \tag{17}$$

where $\chi$ is a constant depending on tectonic and geological conditions. Combination of Eqs. (15), (16), and (17) leads to the log($T$)–$M$ relationship:

$$\log(T)=C+AM \tag{18}$$

where two new parameters are $C=\log\{[a(\alpha\text{-}2)]^{1/\eta}/a(\alpha\text{-}1)\}+[\log(\lambda)-\chi/2]/\eta$ and $A=-1/4\eta$. Obviously, $A$ is positive due to $\eta<0$ because of $1<\alpha<2$ as mentioned above. This results in a positive correlation between $T$ and $M$. When $T$ is known, the value of $M$ for the forthcoming earthquake may be evaluated from Eq. (18), i.e., $M=[\log(T)-C]/A$. From past studies (cf. Wang, 2021b, 2023; and cited references therein), the values of $A$ from observations are all smaller than 1. This leads to $\alpha<1.8$ and thus the values of $\alpha$ for natural earthquakes could be in the range 1.0 to 1.8.

### 3.3 Predicting the Location of a Forthcoming Earthquake

As mentioned by Aki (1989), the earthquake scientists should provide the location of the forthcoming earthquake to the public. Hence, predicting the potential location of the forthcoming earthquake is also important for seismic hazard mitigation. When the stations on which the pre-seismic strains are observed are close to a known active fault, it is very possible to assess the occurrence of the forthcoming earthquake along the fault. On the other hand, when the station site is not close to a known active fault or within a complicated active fault system, it needs other precursors, for example, *b*-value anomalies (e.g., Wang et al., 2016), foreshock activities (e.g., Chen and Wang, 1984; Chen et al., 1990; Gulia and Wiemer 2019; Zaccagnino et al., 2024), geochemical anomalies (e.g., Walia et al., 2009; Fu and Lee, 2018) electromagnetic anomalies (e.g., Ohta et al., 2005, Hayakawa et al., 2006; Hayakawa and Hobara, 2010; De Santis et al., 2019) etc., for helping earthquake scientists to make correct assessment. Hence, researchers have also suggested other methods to judge the possible location of the forthcoming earthquake. Seismologists (e.g., Rundle et al., 2000; Wu et al., 2012) suggested a method to assess the location from seismicity pattern. For some strike-slip and normal earthquakes, seismologists can assess the possible location of the mainshock from its foreshocks (e.g., Chen et al., 1990). Geochemists (e.g., Walia et al., 2009; Fu and Lee, 2018) suggest a method just like that used by seismologists to locate an earthquake from the differences between travel times of *P*- and those of *S*-waves recorded at three stations. They took the occurrence times of geochemical precursors, recorded at three different stations to evaluate the optimal location of a forthcoming earthquake. Geophysicists (e.g., Ohta et al., 2005,

Hayakawa et al., 2006; Hayakawa and Hobara, 2010) suggest the goniometric method to assess the location of the forthcoming event by detecting the directions of ULF emissions from the observational stations to the earthquake epicenter. These methods seem acceptable.

# 4 Discussion

## 4.1 On the Theory for the Pre-seismic Strains

Fig. 1 shows that the strain rate, $\varepsilon_t$, monotonically increases with time. From Fig. 1, Eq. (1) will lead to an increase in the strain acceleration, $\varepsilon_{tt}$, with time. For the time variation as displayed in Fig. 1, at a certain time instant, larger $\alpha$ yields higher $\varepsilon_t$. Meanwhile, there are two steps more or less separated at a particular time instant, $t_c$, which is shorter than $t_f$ and not displayed in the figure. The two steps are: $\varepsilon_t$ first slowly with time when $t<t_c$ and then rapidly with time when $t>t_c$. Such a particular time appears earlier for large $\alpha$ than for small $\alpha$. The second step is the existence of accelerating strain before a forthcoming earthquake from the theoretical studies by Main (1998). From observations of foreshocks, some authors (e.g., De Santis et al., 2015; and Cianchini et al., 2020) applied the revised accelerated moment release model to foreshocks revealing an acceleration pointing to the mainshock. Their model is similar to the present one. Since there is background noise in practical observations, the anomalous strain rate can be measured only in the second step. Like Fig. 1, Fig. 4 also illustrates the similar time variation in the strain, $\varepsilon$. For all cases in Fig. 4, there are also two steps separated at a particular time instant, $t_c$: $\varepsilon$ first slowly with time when $t<t_c$ and then rapidly with time when $t>t_c$. Unlike Fig. 1, such a particular time is almost the same for all $\alpha$'s in use. Meanwhile, in Fig. 4 $\varepsilon$ increases with $\alpha$ when $t$ is smaller than such a particular time; while $\varepsilon$ decreases with increasing $\alpha$, when $t$ is larger than such a particular time. This is the main difference between Fig. 1 and Fig. 4. In addition, larger $\alpha$ produces lower $\varepsilon_t$ as $t$ is approaching $t_f$ in Fig. 4. This means that the strain during a forthcoming earthquake increases with decreasing $\alpha$.

The theory of predicting the failure time of a forthcoming earthquake proposed by this study is basically similar to that used by Das and Scholz (1981) based the Charles

law and that suggested by Main (1988) based on the Voight equation. One difference between this method and theirs is that the values of strain rate at three time instants are taken in this study, while only those of pre-slip at two time instants were considered in theirs. This is due to a reason that they assumed that the model parameters of either Charles law or Voight's equation have been already known, while those in this study are originally unknown and must be estimated from the observations.

Equation (18) exhibits the log($T$)–$M$ relationship based on pre-seismic strains. Tsubokawa (1969, 1973) first obtained a linear relation between the precursor time of crustal movement and mainshock magnitude for Japanese earthquakes in the form: log($T$)=-1.88+0.79$M$, with $C$=-1.88 and $A$=0.79. His observations somewhat confirm the existence of the log($T$)–$M$ relationship. This makes us capable of predicting the magnitude of a forthcoming earthquake when the precursor time has been evaluated from observations. Although the earthquakes used by Tsubokawa (1969, 1973) occurred on different fault zones, his log($T$)–$M$ relationship with the values of $C$ and $A$ represents the average characteristics of crustal deformations in Japan. In general, the parameters $a$ and $\alpha$ of Voight's equation and $\lambda$ and $\chi$ of the scaling laws of faults vary from area to area. Hence, the log($T$)–$M$ relationships might be distinct in different fault systems.

Wang (2023) correlated the precursor time to the earthquake energy. The Gutenberg-Richter's energy-magnitude law of earthquakes (Gutenberg and Richter, 1942, 1956) is: $\log(E_s)=11.8+1.5M$ in which $E_s$ is the seismic-wave energy (in ergs) and $M$ is commonly the surface-wave magnitude, $M_s$. From the law, he obtained the correlation: $M\sim(2/3)\log(E_s)$. In addition, from $\log(T)=C+AM$ he got $\log(T)\sim AM\sim(2A/3)\log(E_s)$. Since $E_s=\xi\Delta E$ where $\Delta E$ is the strain energy of an earthquake and $\xi$ (<1) is the seismic efficiency, Wang (2004) obtained $T\sim\Delta E^{A\xi/3}$. This indicates that the precursor time is dependent on the strain energy of the forthcoming earthquake. The seismic efficiency that depends on the physical and chemical properties of the fault-zone rocks (Knopoff, 1958; Kanamori and Heaton, 2000; Wang, 2009) may also influence $T$. A high seismic efficient will yield a long precursor time.

**4.2 Application of the Theory to Other Earthquake Precursors**

#### 4.2.1 The log(T)−M relationships for other Precursors

In order to measure the pre-seismic strains, the strain-meters should be installed on or much near the fault. When a strain-meter has not installed on or near the fault on which a forthcoming earthquake will happen, it is hence necessary to use other kinds of precursors which are directly or indirectly caused by the pre-seismic fault slip or strains for predicting the earthquake. In other word, it is much significant to explore the application of the present theory on the prediction of $t_f$ and $M$ of a forthcoming earthquake based on other kinds of precursors in practice. The present theory can be applied to other kinds of precursors, and thus the log($T$)−$M$ relationships exist for these precursors. It is significant to apply the above-mentioned theory to predict the failure time and magnitude of a forthcoming earthquake based on other kinds of precursors.

The log($T$)−$M$ relationships have been recognized from the observations of different kinds of precursors for a long time (Rikitake 1975a; Wang, 2021a,b, 2023; and cited references therein). From the plot of $T$ (in days) versus $M$ for five precursors, i.e., crustal movements, electric resistivity, radon (denoted as Rn hereafter) emission, $v_p/v_s$ anomaly, and $b$-value of Gutenberg-Richter frequency-magnitude law (Gutenberg and Richter, 1944). From 30 world-wide earthquakes, Scholz et al. (1973) inferred a relationship: $M_s$=-5.81+1.55log($T$) ($T$ in days) or log($T$)=3.75+0.65$M_s$. For the precursors of crustal deformations and seismic-wave velocities, Whitcomb *et al.* (1973) obtained log($T$)=-1.92+0.80$M_s$ ($T$ in days). Rikitake (1975b) obtained log($T$)= -1.83+0.76$M_s$ ($T$ in days). He also stressed that the log($T$)−$M_s$ relationships are different for different groups of precursors. Rikitake (1979, 1984) divided a large data set of 391 cases of precursors into three classes. He obtained log($T$)=-1.01+0.60$M_s$ for the first class including 192 cases and log($T$)=-1.0 for the second class. He did not report any relationship for the third class for foreshocks, tilt and strain, and earth's currents. Smith (1981, 1986) obtained the following relationship: log($T$)=1.42+0.30$M_s$ ($T$ in years) from the data of abnormal $b$-values for earthquakes in New Zealand. Ding et al. (1985) obtained log($T$)=-0.34+0.38$M_s$ ($T$ in years) for various precursors proceeding large Chinese earthquakes. From the $b$-value anomalies for 45 world-wide earthquakes with 3≤$M_s$≤9, Wang et al. (2016) obtained log($T$)=(2.02±0.49)+(0.15± 0.07)$M_s$ ($T$ in years).

From the previous description, it is clear that the log($T$)−$M$ relationships are

different for distinct kinds of precursors and also region-dependent. These results strongly suggest regional-dependence of *C* and *A* of Eq. (18). Clearly, *C* is influenced by several parameters, while *A* is controlled only by the scaling exponent, $\alpha$, of the fault-zone materials. Hence, *A* is an important indicator of the relationship. The previous studies lead to two interesting points. First, for the same forthcoming earthquake, different kinds of precursors may have different precursor times due to distinct values of *C*, but the same value of *A*. Secondly, for the forthcoming earthquakes that have the same magnitude and occur at different fault zones, different kinds of precursors may have different precursor times due to distinct values of both *C* and *A*.

We will explore the theoretical basis for two kinds of precursors in the followings. The first kind of precursors is the geoelectric signals which are yielded almost within the fault zone where the forthcoming earthquake will happen, and the other is the geochemical signals which might occur on the sites that are somewhat far away from the fault zone. The mechanisms to generate the two kinds of signals will be described below.

#### 4.2.2 For the Geoelectric Precursors

Changes or anomalies of geoelectric signals have been observed prior to earthquakes for a long time (cf. Hayakawa and Hobara, 2010; and cited references therein). Geoelectric signals are associated with pre-seismic slip on a fault where a forthcoming earthquake will happen. It is necessary to build up a comprehensive model that presents the lithosphere-ocean-atmosphere-ionosphere–magnetosphere coupling to interpret the generation of geoelectric precursors (Potirakis et al., 2017; Ouzounov et al., 2018; and cited references therein). Several proposed models are: (1) a model to present Rn ionization and charged aerosol and change of load resistance in the global electric circuit (Ouzounov et al., 2018; Pulinets and Ouzounov, 2018; and cited references therein); (2) a model to show coupling between stressed rocks and the atmosphere–ionosphere system (e.g., Kuo et al., 2011, 2014) based on experimental results of stress-induced charges made by Freund (2002); (3) a model to display ionosphere dynamics with imposed zonal (west-east) electric field (Zolotov et al., 2011, 2012; Namgaladze et al., 2012); and (4) a model of leakage of electric currents

from ocean into the crust having low electric resistivity (Madden and Mackie, 1996). The existence of electric charges/currents on the Earth's ground or in the uppermost crust is a necessary condition for these models. Several mechanisms, including microfracturing (e.g., Ogawa et al., 1985; Molchanov and Hayakawa, 1995), electrokinetic effect (e.g., Mizutani et al., 1976), streaming potentials (e.g., Bernard, 1992), piezoelectricity (e.g., Bishop, 1981; Sornette, 2001; Wang, 2021c), triboelectricity/triboluminescence (e.g., Yoshida et al., *1998),* confined pressure changes (e.g., Fujinawa et al., 2002), the peroxy defect theory (Freund, 2002), piezomagnetism (e.g., Sasai, 1979, 1980; Martin, 1980), etc. have been proposed to explain electric charge generation within the fault zones.

Here, we show three examples to show the geoelectric and geomagnetic precursors caused by pre- seismic ground electric currents. First, Whitworth (1975) proposed a model of the motion of charged edge location (MCD). According to the MCD model, numerous authors (e.g., Tzanis and Vallianatos, 2002; Venegas-Aravena et al.**,** 2019) assumed that an electric current density, $J$, generated within rocks under compressional stress changes with time, i.e., $\sigma_t=\mathrm{d}\sigma/\mathrm{d}t$, can be represented by $J=2^{1/2}(q/\psi B_v)(\sigma_t/Y)$ where $q$ is the linear charge density of edge dislocation, $B_v$ is the Burgers vector module, $\psi$ (=1–3), which represents the dislocation number created by compression and uniaxial tension within a rock (Whitworth, 1975; Vaillianatos and Tzanis, 1998), and $Y$ is the Young's effective module (Turcotte et al., 2003). Since the quantity $\sigma_t/Y$ may be replaced by the strain rate $\varepsilon_t$, the electric current density becomes $J=2^{1/2}(q/\psi B_v)\varepsilon_t$. The geoelectric field is $E=J/\theta_c$, where $\theta_c$ is the electric conductivity, from the Maxwell equation. Meanwhile, the geomagnetic filed at a distance, $r$, from the electric current density is $|B|=\mu_B|J|/2\pi r$, where $\mu_B$ is the permeability of free space, from the Biot-Savart law (cf. Corson and Lorrain, 1962). Clearly, $E$ and $B$ are both related to $\varepsilon_t$. Secondly, Enomoto (2012) obtained $\log(J)=0.5M+\log(5.1\times10^2 eknh^2 D_c/\nu)$ ($e$=the electronic charge; $k$=a constant of proportionality; $n$=the density of negatively charged gas molecules; $h$=the crack gap; $D_c$=critical depth; and $\nu$=the gas viscosity). This shows the correlation between $J$ and $\varepsilon$. Thirdly, some authors (e.g., Sornette, 2001; Wang, 2021c) studied the dependence of ground electric field, $E$, on pre-seismic slip, $u$, in a fault zone in a one-dimensional model with the spatial coordinate $x$ based on the piezoelectricity and the Maxwell's equations. The result is: $E=-\mathrm{i}(c/v)^2(\kappa/\zeta)u$ where $\mathrm{i}=(-1)^{1/2}$ is the imaginary number,

$v=(\mu/\rho)^{1/2}$ is the elastic wave velocity, $\rho$ is the density (kg/m$^3$) of fault-zone rocks, and $c$ is the light speed (=2.999×10$^8$ m/sec in free space), $\zeta$ is the piezoelectric coupling coefficient between elastic field and electric field ($\zeta$=~2×10$^{-12}$ coulomb/ newton for quartz), and $\kappa$ is the wavenumber. Let $L_o$ be the original length of a fault, thus leading to $E=-i(c/v)^2(\kappa/\zeta)(u/L_o)L_o=-i(c/v)^2(\kappa L_o/\zeta)\varepsilon$. The three examples of geoelectric and geomagn anomalies, thus leading to precursors of earthquakes. The precursor times of GEM precursors should be the same as that of the pre-seismic strains. However, Wang (2021a,b) reported different precursor times of electric field and magnetic field even though they appeared before the same earthquake. It is necessary to explore the reasons to cause such a difference in future.

The MCD model is put into the present theory to predict the failure time and magnitude of a forthcoming earthquake. Inserting $E_{tj}$ and $t_j$ ($j$=1, 2, and 3) into Eq. (6) yields

$$E_{tj}=F[a(\alpha-1)](t_f-t_j)]^{(1-\alpha)} \quad (j=1, 2, 3). \tag{19}$$

This leads to

$$t_f=t_j+(E_{tj}/F)^{(1-\alpha)}/a(\alpha-1) \quad (j=1, 2, 3). \tag{20}$$

From Eq. (20), we may predict the failure time, $t_f$, of the forthcoming earthquake. Since $E$ increases with $\varepsilon_t$, their precursor times are the same and thus the precursor time, $T$, is $t_f$-$t_o$. Theoretically, the precursor time of the pre-seismic geoelectric precursor is the same as that of the pre-seismic fault strains. From $T$, we may predict the magnitude of the forthcoming earthquake from Eq. (18), i.e., $M$= [log($T$)-$C$]/$A$.

In principle, the theory works well to predict the failure time of a forthcoming earthquake by using the pre-seismic geoelectric signals. But, in practice there might be a problem that the values of $E_i$ cannot be observed accurately because of the presence of unexpected noise due to thunderstorm, atmospheric abnormal phenomena, and artificial effects. This problem should be very serious when $t<t_c$ because their values are very small and cannot be observed. Hence, the observed data of geoelectric signals must be carefully selected and corrected to remove noise. The visible geoelectric signals should appear when $t>t_c$ because the signals are strong enough. In

addition, in principle $E_i$ must be measured near the fault. But, the monitoring station of geoelectric signals is usually not located near a fault where a forthcoming earthquake will happen. The value of $E_i$ measured at a station not close to the epicenter should be slightly different from and weaker than near-fault one due to attenuation. Nevertheless, the attenuation of geoelectric signals measured at several time instants should be the same on the same station unless there are thunderstorm and abnormal atmospheric phenomena between two time instants of different stations.

#### 4.2.3 For the Geochemical Precursors

Numerous geochemical precursors are not observed at the localities near the earthquake epicenters (Wang 2021a,b; and cited references therein) because the observation stations are not installed at the sites near the epicenters. For example, Rn concentration anomalies prior to an earthquake are often observed somewhat far away from the epicenters because the measurement instruments are installed at hot-water springs or water-wells which may be far away from the epicenters. Nevertheless, their appearances are still related to the pre-seismic slip in the fault zones of forthcoming events. We assume that the presence of Rn concentration anomalies in the underground water might be associated with the spatial distribution of focal mechanism of an earthquake. The spatial pattern of the fault mechanism of an earthquake has four quarters: two for tension or dilatation and others for compression.

Kuo et al. (2010, 2019) reported a positive correlation between the temporal variation in Rn concentrations and that of dilatational strains measured at the Antong station for three events in southeastern Taiwan. The dilatational strains were related to tensional quarters of focal mechanisms of the events as mentioned above. They considered a model to explain Rn volatilization in an undrained fractured aquifer. This model is simply described below. A small fractured aquifer situated in a brittle rock, which is surrounded by a ductile formation in undrained conditions. When aquifer recharge is weak and negligible, undrained conditions are valid. There is only a single water phase in the aquifer before any precursory geochemical phenomenon appears. When the regional stress increases, dilation of brittle rock could occur at a faster rate than the rate of groundwater recharging into the newly created micro-cracks. As a result, gas saturation and two phases (gas and water) develop in the aquifer. The radon in groundwater volatilizes into the gas phase and the Rn concentration in groundwater

decreases. The model is mathematically represented by the following equation:

$$C_w/C_o=(HS_g+1)^{-1} \quad (21)$$

where $C_o$ is the initial Rn concentration (in pCi/L) in formation brine (salt water); $C_w$ is the equilibrium Rn concentration (in pCi/L) remaining in ground-water; $S_g$ is the gas saturation (in %); $H$ is Henry's coefficient (dimensionless) for Rn. From the rock-dilatancy model (Brace et al., 1966): $\varepsilon_v=S_g/(1/\phi)$ or $S_g=\varepsilon_v/\phi$ where $\varepsilon_v$ and $\phi$ denote, respectively, the (dimensionless) volumetric strain of the rocks beneath the observation site and the initial fracture porosity before rock dilatancy. The volumetric strain may be represented as $\varepsilon_1+\varepsilon_2+\varepsilon_3$ where $\varepsilon_j$ is the strain along the $j$-th axis ($j$=1, 2, and 3) (Turcotte and Schubert, 1982). This yields $S_g=(\varepsilon_1+\varepsilon_2+\varepsilon_3)/\phi$. Equation (19) shows that $C_w$ increases with decreasing $S_g$. Inside the brittle rocks underneath the observation site, $S_g$ increases with $\varepsilon_v$, thus leading to a decrease in $C_w$. The value of $\varepsilon_v$ inside the brittle rocks underneath the observation site will be induced by the strain in the fault zone where the forthcoming earthquake will occur. Hence, the Rn concentration changes are controlled by pre-seismic strains that occur in the related fault zone.

Note that although we have considered a model to describe the production of preseismic geochemical signals, the production processes could be more complicated than the present model. Schirripa Spagnolo et al. (2024) addressed that preseismic geochemical signal are produced by the transport of chemical markers throughout the aquifers producing complex spatial circulations and alterations which can be extremely difficult to grasp using just one single model. They also claimed that such complex interactions among fault zones, host rocks upper and lower crustal volumes produce a wide range feedback mechanisms. These problems are beyond the scope of this study and need further investigations.

Of course, the time-dependent pre-seismic slip or strain on a fault along which a forthcoming earthquake will happen can produce stress changes surrounding the fault (Aki and Richards, 1980). This might induce some geochemical precursors which occur on some places somewhat far away from the fault. Hence, such kinds of precursors will appear more or less later than the pre-seismic slip or strain that happened on the fault. This results in a shorter precursor time than that for the

pre-seismic slip or strain. Here, we consider a mechanical model to explain the problem. Dobrovolsky et al. (1979) used a half space, during the preparation processes of an earthquake, a zone of cracked rocks is formed in the focal area under the tectonic loading, $\tau$. The media inside the zone may be considered as a solid inclusion with different moduli that are lower than that of the half space. The solid inclusion re-distributes the stresses accompanied by deformations, including those on the Earth's ground surface. Let $V$ be the solid soft inclusion volume that is an ellipse with a long-axis length of $l_l$ and a short-axis length of $l_s$: $l_l$>l for $M\geq5$ and $l_l=l_s$ for $M<5$, thus leading to $V=\pi l_l l_s^2/6$ for $M\geq5$ and $V=\pi l_s^3/6$ for $M<5$. The shear modulus of the half space and that of the inclusion are $\mu$ and $\mu$-$\delta\mu$, respectively. The ratio $\delta\mu/\mu$ is denoted by $\varphi$. Assuming that the zone of effective manifestation of the precursory deformations is a sphere with the center at the epicenter of the forthcoming earthquake under the shear stresses loaded at infinity. In the spherical zone with a radius of $r_\varepsilon$, the deformation has a strain being equal to or exceeding a certain $\varepsilon_s$ which is smaller than the strain on the related fault. The $r_\varepsilon$ is called the 'strain radius.' They obtained $r_\varepsilon=0.85(\varphi V\tau/\mu\varepsilon_s)^{1/3}$. This leads to

$$\varepsilon_s=(0.85)^3\varphi V\tau/\mu r_\varepsilon^3. \qquad (22)$$

This reveals that the strain decreases when the radius or the distance from the earthquake hypocenter increases. Based on Eq. (22), Rn concentration anomaly could occur at a distance $r_\varepsilon$ from the hypocenter when the strain at the observation site is larger than $\varepsilon_s$. Hence, the pre-seismic strain in the related fault zone must be larger than a particular value, $\varepsilon_p$ (>$\varepsilon_o$), at time $t=t_p$. This makes the occurrence time of Rn concentration anomaly be later than that of the pre-seismic strain because of $t_p>t_o$. Thus, the precursor time of the former is shorter than that of the latter. Equation (5) becomes

$$\varepsilon(t)-\varepsilon_p=\{[a(\alpha\text{-l})(t_f-t_p)+\varepsilon_{tf}^{\ 1-\alpha}]^{\eta}-[a(\alpha\text{-l})(t_f-t)+\varepsilon_{tf}^{\ 1-\alpha}]^{\eta}\}/a(\alpha-2). \qquad (23)$$

Define $T=t_f-t_p$ to be the precursor time of this precursor. Considering $\varepsilon_p=\gamma\varepsilon_f$ and $\varepsilon_{tf}$>>1, Eq. (23) hence becomes

$$(1-\gamma)\varepsilon_f=\{[a(\alpha\text{-l})T]^{\eta}/a(\alpha-2). \quad (24)$$

This yields

$$T=[a(\alpha-2)(1-\gamma\varepsilon_f)]^{1/\eta}/a(\alpha\text{-l}). \quad (25)$$

Taking the logarithm or the two sides of Eq. (25) leads to

$$\log(T)=[a(\alpha-2)(1-\gamma)\varepsilon_f]^{1/\eta}/a(\alpha\text{-l}). \quad (26)$$

This gives

$$\log(T)=C^{'}+AM_w \quad (27)$$

where $C^{'}$=(1-$\gamma$)$C$<$C$. This indicates that when the Rn concentration anomaly is taken as a precursor, only the value of the constant is reduced from $C$ to $C^{'}$, while the scaling exponent $A$ does not change because of the same fault zone. This again to confirm the importance of the log($T$)–$M$ relationship on the assessment of a forthcoming earthquake. When two groups of earthquakes occur in two fault systems whose rock materials have different values of $a$ and $\alpha$, their values of $C$ and $A$ could be different, thus resulting in different log($T$)–$M$ relationships.

For Rn concentration anomalies before six earthquakes with $M$=5.0−6.8 and $d$=7.0−35.6 km ($M$=the local magnitude; $d$=the focal depth, in km) in southeastern Taiwan, Kuo et al. (2020) obtained log($T$)=1.456+0.053$M$. For the Rn concentration anomalies before 9 events in northern Taiwan, Wang (2023) obtained log($T$)=(-0.21±0.30)+(0.23±0.02)$M$. For the Rn concentration anomalies before 111 earthquakes in Taiwan, Wang (2021b) obtained log($T$)=(-2.05±0.40)+(0.58±0.01)$M$ for the events with $d$≤40 km and $\Delta$≤40 km ($\Delta$=the focal depth, in km); and log($T$)=(-0.40±0.42)+(0.26±0.01)$M$ for those with $d$>40 km or $\Delta$>40 km. The log($T$)−$M$ relationship for northern Taiwan is different from that for southeastern Taiwan. This indicates the difference on $a$ of the fault-zone rocks between the two areas. The log($T$)−$M$ relationship for northern Taiwan is different from those for Taiwan in two different focal-depth ranges. This suggests that there is a difference on $\alpha$ of the

fault-zone rocks between northern Taiwan and the whole Taiwan region. That the $\log(T)-M$ relationships for Taiwan in two different focal-depth ranges suggests that the fault-zone rocks in the two different focal-depth ranges are different from each other.

We assume that the theory proposed in this study can be applied to other kinds of precursors, and thus the $\log(T)-M$ relationships exist for these precursors as mentioned above. Based on the difference of the $\log(T)-M$ relationships between two kinds of precursors, Wang (2023) suggested a method to predict the failure time and magnitude of a forthcoming earthquake directly from observations. He explored in details the conditions of the values of *C'* and *A* of Eq. (25) for two different precursors that can be used for earthquake prediction. He also gave examples for geochemical precursors to show how to predict the failure time and magnitude of a forthcoming mainshock. The present theory provides the physical basis of his study.

## 5. Conclusions

From the subcritical crack growth model, we propose a theory of predicting a forthcoming earthquake from pre-seismic strain signals. We consider three aspects: prediction of failure time, prediction of earthquake magnitude, and prediction of location. The pre-seismic strain is here considered as a fundamental and important earthquake precursor. Based on the Voight's equation for failure of materials under stresses, we theoretically investigate the physical basis on predicting the failure time and magnitude of a forthcoming earthquake in terms of pre-seismic anomalous strain signals which are generated on or near the fault where the event will happen. Meanwhile, the present study demonstrates the physical basis of the $\log(T)-M$ relationships of precursors. Results exhibit that the failure time depends on the strain rate and two parameters of the Voight's equation; while the magnitude are controlled by the precursor time, two parameters of the Voight's equation, and the exponent of the scaling law between the co-seismic strain and the fault length. The scaling exponent, $\alpha$, of the Voight's equation is an important factor on the $\log(T)-M$ relationship. Although the location of a forthcoming earthquake cannot be determined from the present theory, it may still be qualitatively assessed from the observations. The theory may be applied to the $\log(T)-M$ relationships of other kinds of precursors.

Based on the theoretical results made by Main (1998) and the observed values of *A* of the relationships, the value of $\alpha$ must be in the range 1.0 to 1.8 for the generation of earthquakes. The log($T$)−$M$ relationships of pre-seismic geoelectromagnetic and geochemical signals are taken into account. Theoretical results reveal that the precursor times of the pre-seismic geoelectromagnetic precursors and those of geochemical precursors are, respectively, the same and shorter than that of the pre-seismic strains.

*Data availability*. No

*Competing interests*. There are no known competing financial interests or personal relationships that could have appeared to influence the work reported in this paper.

*Acknowledgments*. This study was supported by the Institute of Earth Sciences, Academia Sinica, Taiwan, ROC.

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

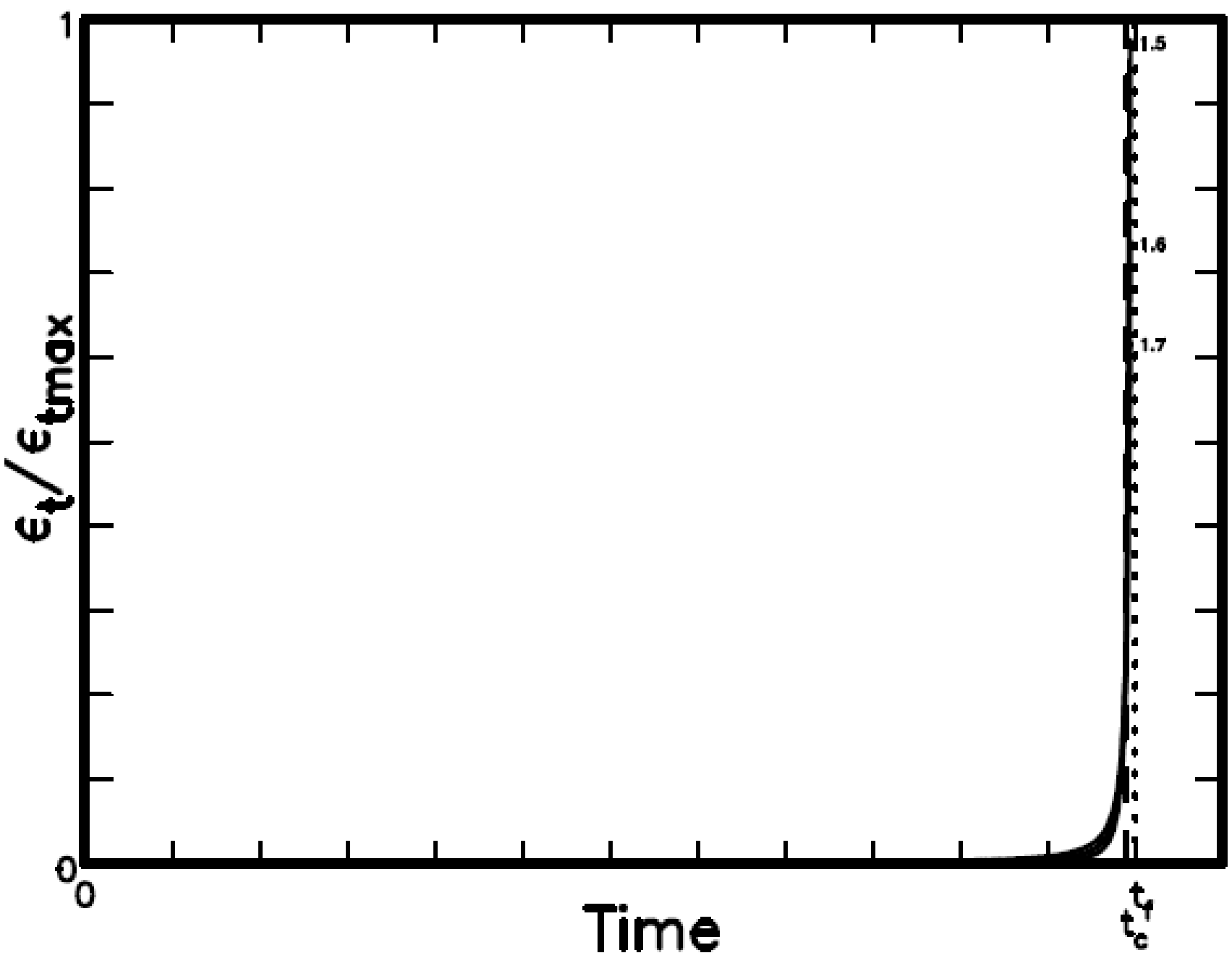

Figure 1. The plot shows the time variations in strain rate, $\varepsilon_t(t)$, for $\alpha$=1.5, 1.6, and 1.7 when $a$=0.5. The three curves intersect one another at the point with $t$=$t_c$.

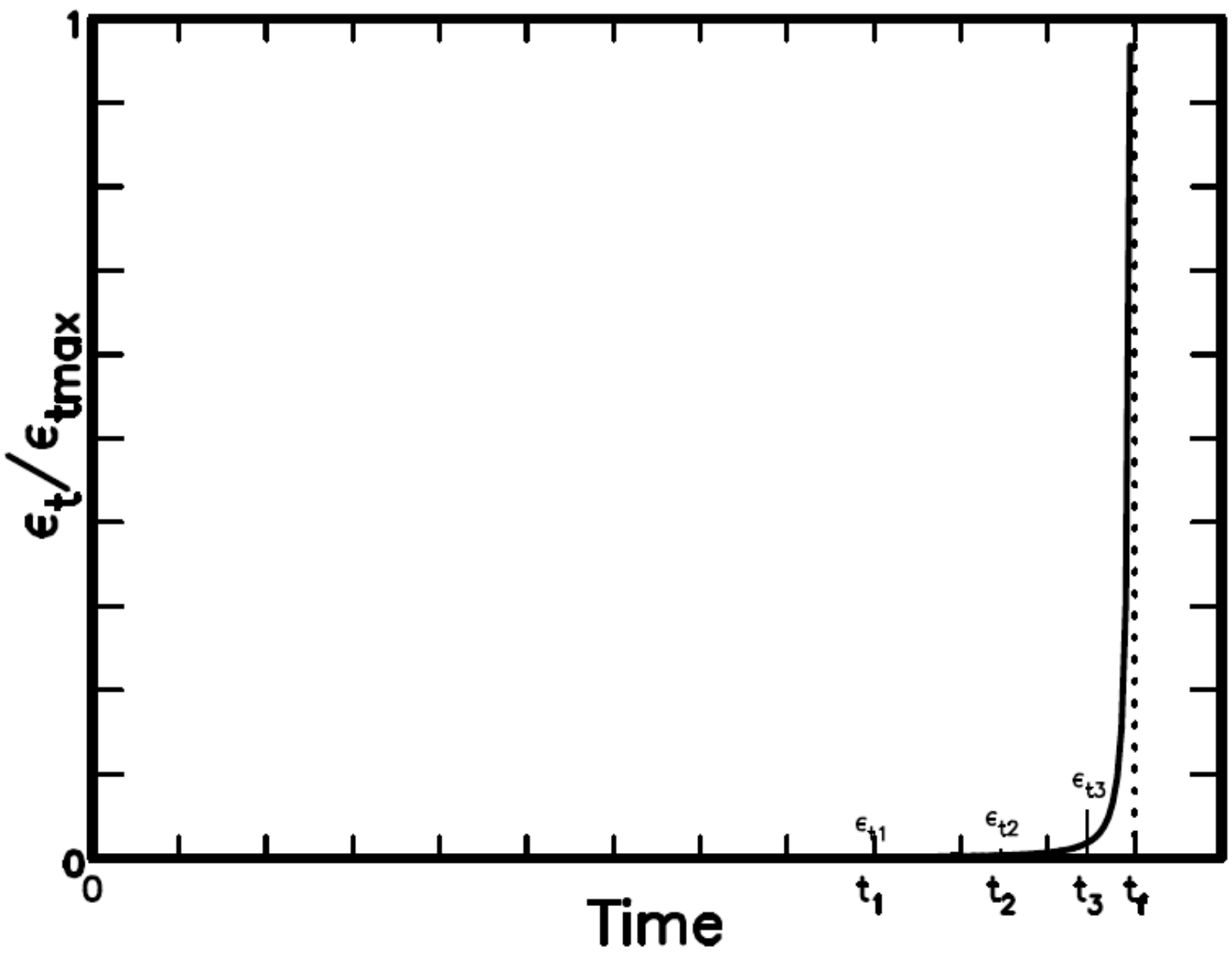

Figure 2. The plot shows the time variation in strain rate, $\varepsilon_t(t)$, and three values of $\varepsilon_t(t)$, i.e., $\varepsilon_{t2}$, $\varepsilon_{t2}$, and $\varepsilon_{t3}$, at three time instants, $t_1$, $t_3$, and $t_3$ for $\alpha$=1.6 when $a$=0.5.

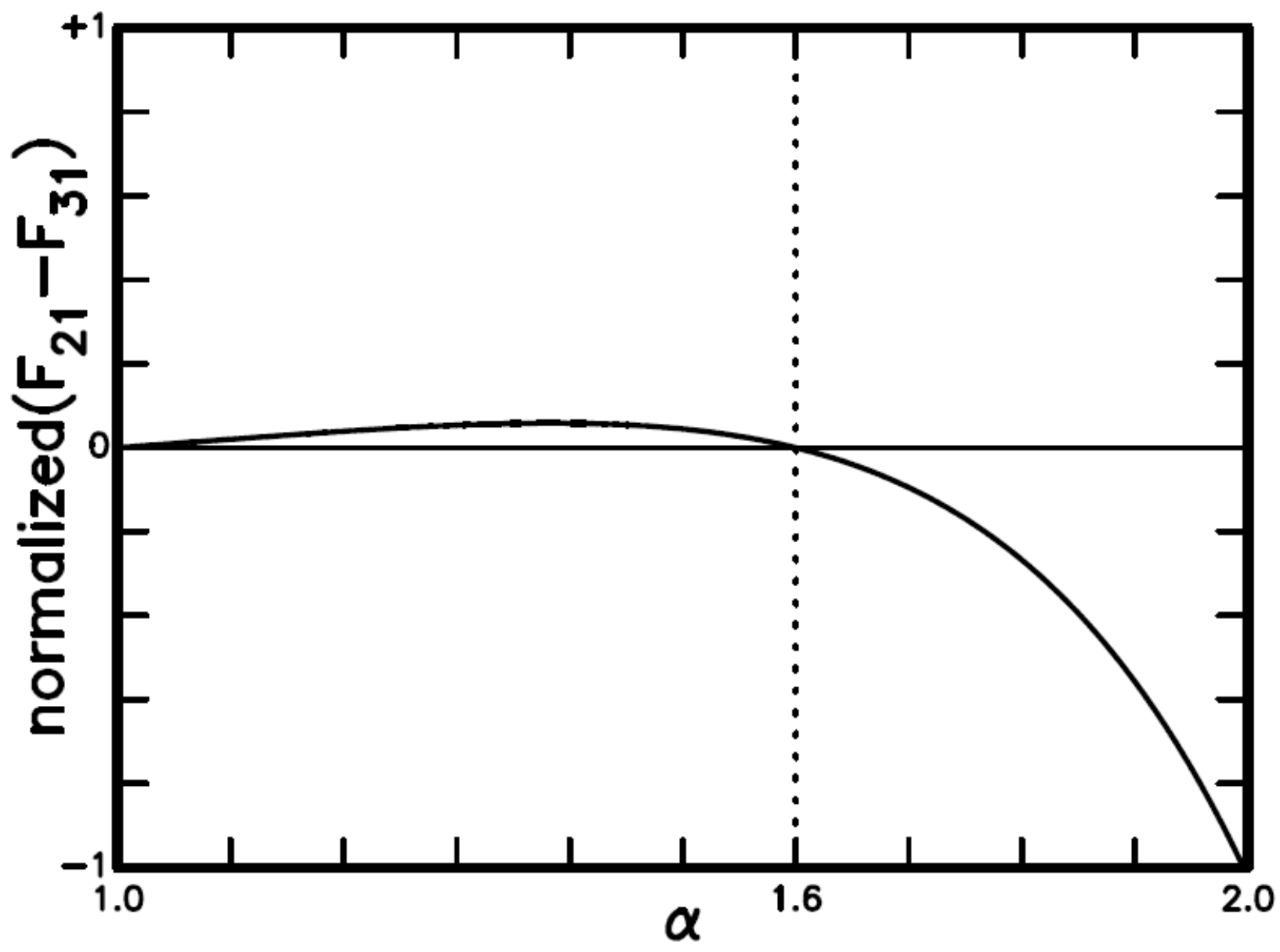

Figure 3. The plot displays the curve for $F_{21}(\alpha)$-$F_{31}(\alpha)$. The intersection point of the curve and the line with $F_{21}(\alpha)$-$F_{31}(\alpha)=0$ is at $\alpha=1.6$.

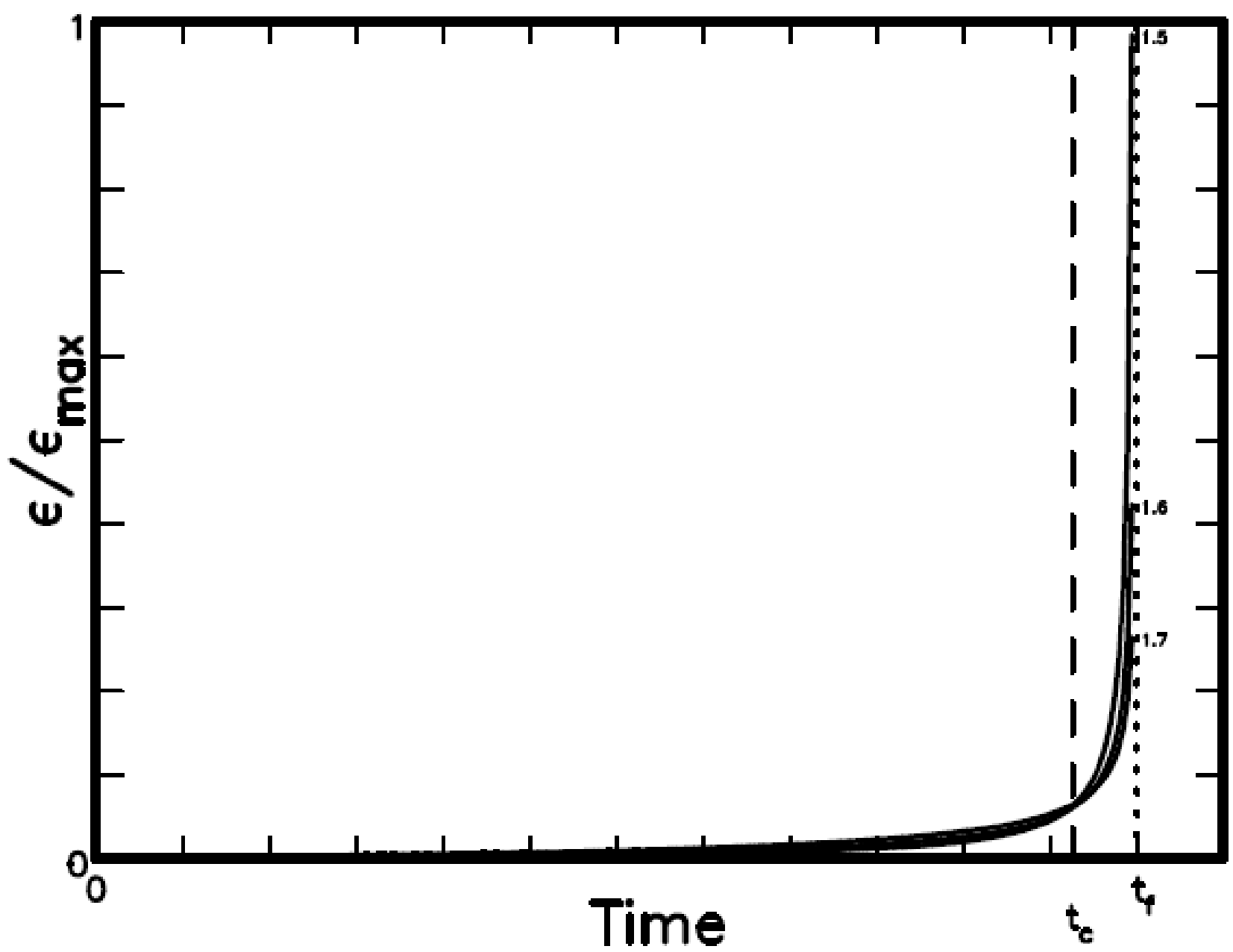

Figure 4. The plot shows the time variations in strain, $\varepsilon(t)$, for $\alpha$=1.5, 1.6, and 1.7 when $a$=0.5. The three curves intersect one another at the point with $t$=$t_c$.