# Peer review of "A Theory of Earthquake Prediction"

_EGUsphere, 2025_

## Referee Comment (RC1)

Title: A Theory of Earthquake Prediction
Author(s): Wang, J.-H.
Article reference: DOI:10.5194/egusphere-2025-3192
Referee comments

The manuscript discusses a theory to predict the time to failure, moment, and location of the earthquake by monitoring precursory signals of strain rate increase in Earth's crust before the earthquake. The author also discusses various geochemical and geoelectric signals associated with the mechanical deformation of rocks due to the increase in the strain rate, which can be monitored as precursors before the earthquake.

I would like to recommend **rejecting** the manuscript for publication in its current version due to conceptual errors in reasoning. The detailed response is as follows:

The manuscript discuss three main results regarding time to failure, moment and location of an earthquake in Sec. 3.

(i) Time to failure (Sec 3.1): The author assumes a power law scaling of strain rate simplifying Eq. 4 (which is based on Voigt equation or quasi-static crack growth theory),

$$\dot{\epsilon}(t) \sim (t - t_f)^{1/(1-\alpha)},$$

by making assumption that magnitude of strain rate at failure time $\dot{\epsilon}(t_f)$ will be much larger than "1 strain /sec". This statement is simply wrong. The correct statement is

$$\dot{\epsilon}(t_f) \gg [a(\alpha - 1)t_f]^{1/(1-\alpha)},$$

which means it depends of $a$ and $\alpha$, the fitting parameters of the above power law.

Further, Eq. 8-11 is just explaining a way to obtain three parameters of model $\dot{\epsilon}(t_f)$, $a$, and $\alpha$ by fitting three data points. In natural observations, these precursory "quasi-static" strain rates can be monitored by GPS stations using ground deformations, which typically have a reading per day. This means these precursory signals need to be fitted against much more dense data, and elaborate methods will be required for that.

Apart from that, the Figures 1-4 used to explain these equations are shown without any information of units or using non-nondimensionalization. At least they should be shown on a log-scale to clearly see $t_c$, $t_f$, etc.

(ii) Moment of earthquake (Sec 3.2): (ii) Moment of earthquake (Sec 3.2): The author argues that the strain at time of failure $\epsilon(t_f) = \epsilon_f$ can be considered as average strain after failure (or an earthquake) as the duration of the earthquake is small. *This statement is completely wrong.* The co-seismic deformation during an earthquake is much larger than any "quasi-static" deformation during the precursor phase; therefore, Shaw 2023 scaling (line 305), which relates the co-seismic slip with the rupture length (not fault length), is simply not applicable here. Due to the conceptual mistake in this argument, the main result of this section, i.e. Eq. 18, is not correct, and all other discussions based on it in the rest of the manuscript are highly doubtful.

(iii) Location of earthquake (Sec 3.3): This section does not present any mechanical model or argument regarding the location of an impending earthquake with respect to observation points. A general discussion about how precursory signals can be utilized to locate earthquakes does not constitute a scientific argument that warrants publication.

---

## Author Comment (AC1)

**My Reply to the Comments by Reviewer 1**
Title: A Theory of Earthquake Prediction Author(s): Wang, J.-H. Article reference: DOI:10.5194/egusphere-2025-3192 Referee comments

In reply to the comments given by Reviewer 1, my answers to three main questions are described below.

The manuscript discusses a theory to predict the time to failure, moment, and location of the earthquake by monitoring precursory signals of strain rate increase in Earth's crust before the earthquake. The author also discusses various geochemical and geoelectric signals associated with the mechanical deformation of rocks due to the increase in the strain rate, which can be monitored as precursors before the earthquake. I would like to recommend rejecting the manuscript for publication in its current version due to conceptual errors in reasoning. The detailed response is as follows: The manuscript discuss three main results regarding time to failure, moment and location of an earthquake in Sec. 3.
**[Answer] I am much appreciated with your comments which have helped me to re-think related problems in depth.**

(i)Time to failure (Sec 3.1): The author assumes a power law scaling of strain rate simplifying Eq. 4 (which is based on Voigt equation or quasi-static crack growth theory), $\epsilon'(t) \sim (t\text{-}t_f)^{1/(1-\alpha)}$ , by making assumption that magnitude of strain rate at failure time $\epsilon'(t_f)$ will be much larger than "1 strain /sec". This statement is simply wrong. The correct statement is $\epsilon'(t_f) \gg [a(\alpha\text{-}1)t_f]^{1/(1-\alpha)}$ , which means it depends of a and α, the fitting parameters of the above power law. Further, Eq. 8-11 is just explaining a way to obtain three parameters of model $\epsilon'(t_f)$, a, and α by fitting three data points. In natural observations, these precursory "quasi-static" strain rates can be monitored by GPS stations using ground deformations, which typically have a reading per day. This means these precursory signals need to be fitted against much more dense data, and elaborate methods will be required for that. Apart from that, the Figures 1-4 used to explain these equations are shown without any information of units or using non-nondimensionalization. At least they should be shown on a log-scale to clearly see $t_c$, $t_f$, etc.
**[Answer] From numerous observations, the values of $\alpha$ are usually in between 1 and 2. This can be seen in the text of my manuscript. Hence, the condition $1<\alpha<2$ is considered in the study. For $\alpha>1$, the strain rate is $\varepsilon_t=[a(\alpha\text{-}1)](t_f\text{-}t)+\varepsilon_{tf}^{(1\text{-}\alpha)}]^{1/(1-\alpha)}$, which is Eq. (4), which depends on $a$ and $\alpha$, in my manuscript. Due to $1\text{-}\alpha<0$, $\varepsilon_{tf}^{1\text{-}\alpha}$ is much smaller than 1 strain/sec because the strain rate, $\varepsilon_{tf}$, at the failure time should be much larger than 1 strain/sec, and thus this term can be excluded in Eq. (4). This makes Eq. (4) become $\varepsilon_t=[a(\alpha\text{-}1)](t_f\text{-}t)]^{1/(1-\alpha)}$ (i.e., Eq. (6) in the manuscript) rather than the inequality: $\varepsilon_t \gg [a(\alpha\text{-}1)(t_f\text{-}t)]^{1/(1-\alpha)}$ as claimed by Reviewer 1. This suggests that my assumption could be OK.
In order to compare the variations in strain rate for different values of α, I used the normalized values in Figures 1−4. Actually, it is necessary to plot the real observed values in the practical applications. Of course, a log-scale for t or a log-log scale for ε/ε$_{max}$ versus t is also a good choice.**

(ii) Moment of earthquake (Sec 3.2): (ii) Moment of earthquake (Sec 3.2): The author argues that the strain at time of failure $\epsilon(t_f)=\epsilon_f$ can be considered as average strain

after failure (or an earthquake) as the duration of the earthquake is small. This statement is completely wrong. The co-seismic deformation during an earthquake is much larger than any "quasi-static" deformation during the precursory phase; therefore, Shaw 2023 scaling (line 305), which relates the co-seismic slip with the rupture length (not fault length), is simply not applicable here. Due to the conceptual mistake in this argument, the main result of this section, i.e. Eq. 18, is not correct, and all other discussions based on it in the rest of the manuscript are highly doubtful.

**[Answer] Although the Voight equation is a kind of quasi-static subcritical crack growth theory, but the strain changes from a small value to a big one after $t>t_c$ as displayed in Figure 1. After $t_c$, numerous precursors will appear. At time of failure $\varepsilon(t_f)=\varepsilon_f$ must be very large. My assumption is that $\varepsilon_f$ plays an actor of the source of strain energy of an impending earthquake and thus it can be considered as average strain after failure (or an earthquake) because the duration of the earthquake is usually small. Could you show me some theoretical results to demonstrate the difference, if you cannot accept my assumption? Thanks.**

**On the other hand, the average co-seismic strain, $\varepsilon_{cs}$, should be K times of $\varepsilon_f$ on the basis of your viewpoint, thus leading to $\varepsilon_{cs}=K\varepsilon_f$. This would make Eq. (16) become $\log(K\varepsilon_f)=\log(\lambda)-\log(L)/2$ or $\log(\varepsilon_f)=-\log(K)+\log(\lambda)-\log(L)/2$. This means that we should add $K$ or $\log(K)$ into the related equations after Eq. (16). This makes us be still able to predict the magnitude of an impending earthquake from the present theory. Of course, we should study the value of K in advance. Could you accept my consideration?**

(iii) Location of earthquake (Sec 3.3): This section does not present any mechanical model or argument regarding the location of an impending earthquake with respect to observation points. A general discussion about how precursory signals can be utilized to locate earthquakes does not constitute a scientific argument that warrants publication.

**[Answer] In spite of the case that the sites of strain-meters are near the fault where an earthquake will occur soon, it is not easy to predict the location of an impending earthquake on the basis on the present theory. This is a weak point of my theory. Hence, I introduced some ways to predict the location of an impending earthquake suggested by other earthquake scientists. I hope this will help the readers who are not so familiar with this study area. Could you accept my viewpoint?**

---

## Author Comment (AC2)

Journal reviewer's marked copy.

Reviewer's margin comments refer to yellow-highlighted text.

Blue highlighting was for the reviewer's comprehension, and may be ignored.

**A Theory of Earthquake Prediction**

Jeen-Hwa Wang

- 4 Institute of Earth Sciences, Academia Sinica, Taipei, Taiwan, ROC
- 5 E-mail: jhwang@earth.sinica.edu.tw

**Abstract**

In this study, the pre-seismic strain of an earthquake is considered as a fundamental and important precursor. Based on the Voight's equation for material failure, we theoretically investigate the physical basis on predicting the failure time, magnitude, and location of a forthcoming earthquake in terms of pre-seismic strains generated on or near the related fault where the event will happen. The log(T)-M relationship is built up. Results exhibit that the failure time depends on the strain rate and two parameters of the Voight's equation; while the magnitude is associated with the precursor time, two parameters of the Voight's equation, and the exponent of the scaling law between the strain and the fault length. The location of the forthcoming earthquake may be qualitatively estimated from the localities of observation sites where the pre-seismic strains are observed. In addition, the anomalous geoelectric and

The abstract must stand alone. It is incomprehensible unless variables T and M are defined.

Precursory strain?

Keywords: Earthquake prediction, strain, failure time, magnitude, location, geochemical signals prior to earthquakes are also taken into account as precursors.

Their  $\log(T)$ -M relationships are derived. The precursor times of geoelectric signals and those of the geochemical signals are, respectively, the same and shorter than that

Voight's equation, fault length of the pre-seismic strains.

**1 Introduction**

49

The ruptures of earthquakes, especially for large ones, are usually preceded by complex physical and chemical processes which may produce the so-called precursors (e.g., Atkinson, 1984; Main and Meredith, 1989; Main, 1999; Zaccagnino and Doglioni, 2022). Hence, a significant way to reduce seismic hazards is the prediction of forthcoming earthquakes based on observations of reliable precursors. Since Milne (1880) first addressed this viewpoint in the nineteenth century, earthquake prediction has been a challenging problem for earthquake scientists (e.g., Knopoff, 1996). Aki (1989, 2009) assumed that earthquakes are predictable and earthquake scientists should inform the probability of the occurrence of an earthquake with a specified magnitude, place, and time window to the government and the public for mitigating hazards. Although the earthquake prediction seems successful for few large events, including the 1975 Haicheng, China, earthquake (cf. Wang et al., 2006), it has been long a debatable problem of earthquake science. Numerous earthquake scientists address that earthquakes can be predicted, but some others stand for the opposite viewpoint (e.g. Geller, 1997; Geller et al., 1997). The latters were mainly based on the reasons that the brittle crust is quite disordered and complicated (cf. Savage et al., 2010) and it sometimes exists in the critical state (cf. Bak, 1996). The two conditions will reduce the predictability of forthcoming earthquakes. However, disorder and complexity within a single fault could be much lower than those in the brittle crust or a fault system. A fault could be at the subcritical state (cf. Atkinson, 1984; Main and Meredith, 1989) before its failure occurs. Hence, it is still significant to explore an acceptable, workable model for predicting the failure time,  $t_f$ , the magnitude, M, and the source area of a forthcoming earthquake from observed precursors, especially for a single fault. Although reliable precursors may provide us a clue to judge whether or not an

You will immediately lose a large number of readers, with such an opening sentence. Although some papers (mostly from the 1970's to 1990's) explore possible precursory signals, evidence is variable at best, controversial certainly. It is a huge leap to extend from a few studies of selected earthquakes, to stating that large earthquakes are "usually" preceded by precursory processes and (by implication) by observable precursors.

This begins a more reasonable discussion of the fact that earthquake prediction is debatable. Presenting the topic in a fair and balanced way does not detract from the purpose of the paper.

This is a good point.

earthquake will happen in an area, the observations of precursors that are merely on the reduction side of science (see Kuhn, 1962) thus cannot be directly applied to predict anything. Hence, earthquake scientists need workable theories or models, which are on the deduction side, for prediction. Up to date, the reduction side is much stronger than the deduction one on the earthquake prediction research. This cannot

scientific community in order to advance physical theories and models towards the great goal of earthquake prediction. One of the most important matters is the construction of physico-chemical models for respective precursors or even a are of the "reductionist" unified model for all precursors. Through the comparison between the observations and the models, earthquake scientists could obtain the optimum ones for respective precursors or the optimum unified one. Based on the optimum models or the optimum unified one, earthquake scientists may be capable of predicting an earthquake, including its location, time window, and magnitude as mentioned above. Of course, such a model could be region-dependent, because different tectonic and geological conditions will influence the parameters of the model.

Many of the cited papers type. The author could do a service by pointing this out, or pointing to some examples of this among the many papers he cites. And also to some papers that follow the more-desired deductive approach.

Reid's elastic rebound theory (Reid, 1910) assumes that the loading stress and slip on a fault are the major factors in causing an earthquake rupture. Numerous authors (e.g., Dieterich, 1978; Lomnitz and Lomnitz-Adler 1981; Kostrov and Das, 1982; Main, 1988, 1999; Scholz, 1990) assumed that the pre-seismic stress,  $\sigma$ , and slip, u (or strain,  $\varepsilon$ ), on a fault are two important factors in influencing the generation of precursors. Anomalous pre-seismic displacements or strains near the faults have been observed before numerous earthquakes. Tsubokawa et al. (1964) first measured pre-seismic displacements at several inland sites before the June 16 1964 M7.5 Niigata, Japan, earthquake. Kanamori (1973, 1996) reported pre-seismic release associated with forthcoming major earthquakes, especially in Japan. Yu et al. (2001) reported the pre-seismic displacements on the near-fault stations before the September 20 1999 M7.6 Chi-Chi, Taiwan, earthquake. Papazachos et al. (2002) found accelerating pre-seismic crustal deformation before large earthquakes in the Southern Aegean area. Sarkar (2011) observed possible accelerated Benioff strains prior to large earthquakes in the Sistan Suture Zone of Eastern Iran. These studies confirm the significance and importance of pre-seismic slip or strain on either earthquake prediction or assessment for forthcoming earthquakes. These studies confirm the significance and importance of pre-seismic slip or strain on earthquake prediction or assessment of forthcoming earthquakes. Laboratory experiments reveal that  $\sigma$  and u are time-varying (Atkinson, 1984;

It is disingenuous to neglect to mention that many (more?) studies have failed to identify precursory strains, displacements, or anything else before earthquakes. This sets a biased tone for the entire paper.

These selected studies do not "confirm" the claimed points. They are suggestive, and were among those that prompted a great deal of careful observational work that largely failed to confirm the existence of observable precursors.

By not acknowledging decades of work, large bodies of literature, negative findings, and alternative hypotheses, the author risks relegating this paper to join others in a "fringe" view of earthquake physics, a "truebeliever" category.

Rudnicki, 1988; Main and Meredith, 1989). While, the slip as well as the strain increased very slowly with time from the initial time  $t_0$  to a particular time  $t_c$  and then increased rapidly from  $t_c$  up to the failure time  $t_f$  when an earthquake happens This is

101

103104

118

the so-called quasi-static subcritical crack growth (SCG) model (Atkinson, 1984, 1987; Atkinson and Meredith, 1987) which is usually represented by the Charles law (e.g., Das and Scholtz, 1981; Main, 1988, 1999). Das and Scholz (1981) used this model with Charles law to describe the acceleration of a crack tip from an initially slow (sub-critical) rate due to stress corrosion to rapid remarkable rupture under increasing stresses. They predicted the failure time which depends on initial conditions on a fault, such as crack length, crack-tip velocity, residual frictional stress following a previous earthquake, stress-corrosion index, and the rate of stress input. Main (1988) applied a similar theory to predict the occurrence time of an event. His model may quantitatively explain the decrease of failure time in the crust in terms of decreases in the residual stress due to increasing heat flow, coupled with increases in both stress-input rates and density of nucleation points for rupture initiation. The model also predicts progressively increasing failure times for normal, strike-slip, and thrust faults under similar conditions. Wang (2021a,b; 2023) and Wang et al. (2016) classified the long-term, intermediate-term, short-term, and immediate-term precursors based on the SCG (subcritical crack growth) model as mentioned above.

Rewording needed: Laboratory observations are not a model. Atkinson and Meredith applied the SCG model to the laboratory data.

Which model? (Several have been mentioned in preceding sentences.)

$$X_{tt}-aX_{t}-a=0 (1)$$

rate-dependent law for material failure:

From rock mechanic experiments, Voight (1988, 1989) proposed a nonlinear where X is an observable quantity,  $X_{tt}$  and  $X_{t}$  denote  $d^{2}X/dt^{2}$  and dX/dt, respectively,  $\alpha$  is a constant, and  $\alpha$  is the scaling exponent of the model. Based on rock mechanics, X may be interpreted in terms of conventional geodetic observations (e.g., length change, fault slip, strain or angular change), seismic quantities (e.g., the square root of cumulative energy release or Benioff strain) or geochemical observations (such as gas emission rates or chemical ratios). The parameter  $\alpha$  varies with rock materials and also depends on the temperature. Eq. (1) is called the Voight's equation hereafter. Some authors (e.g., Varnes, 189; Kilburn and Voight, 1998) compared Eq. (1) with the Charles law for the SCG model. Essentially, the Voight's equation is similar to the Charles law. The Voight's equation has been applied to predict the failure time of an earthquake based on the accelerated Benioff strain (e.g., Bufe and Vanus, 1993; Bowman et al., 1996) and the accelerating strain (e.g., Main, 1999). In addition, Main (1999) also studied the failure times of earthquakes by considering constitutive rules of a simple percolation model (e.g., Stauffer and Aharony, 1994). However, they did not predict the magnitude of a forthcoming earthquake.

The pre-seismic strains observed on or near a fault are directly related to the stress and slip on the fault zone. Define  $T=t_ft_0$ , where  $t_0$  is the initial occurrence time of the precursor, be the precursor time (see Wang et al., 2016; Wang, 2021a,b). In this study, we will propose a theory to predict the failure time,  $t_f$ , magnitude, M, and location of a forthcoming earthquake and to investigate the relationship between the precursor time and earthquake magnitude from the pre-seismic and co-seismic strains based on the Voight's equation. In addition, the theory can be also applied to other kinds of precursors.

139 140

128129

**2 Voight's Equation**

141142

From the results obtained from the rock mechanics experiments, Voight (1988) proposed the empirical equation, i.e., the so-called Voight's equation, to describe rate-dependent material failure. The Voight's equation has been considered as a fundamental physical law governing diverse forms of material failures (e.g. Voight, 1988, 1989). It is a more general form of Charles' law (Main, 1999). Like several authors (e.g., Das and Scholtz, 1981; Main, 1988, 1999), I assume that this empirical equation can be applied to real earthquakes. In addition, this empirical equation has been applied to volcanic eruptions (Voight, 1988b; Cornelius and Voight, 1995; Kilburn and Voight, 1998).

If X in Eq. (1) is taken to be the strain,  $\varepsilon$ , on a fault, the final stages of failure under steady conditions of a rock in compression would show a proportionality between the logarithm of creep acceleration and the logarithm of creep velocity. Integrating Eq. (1) gives the expression for the strain rate,  $\varepsilon_t$ , and strain acceleration,  $\varepsilon_t$ , on a fault zone. In the followings, the strain and strain rate at the initial time,  $t_o$ , are denoted by  $\varepsilon_o$  and  $\varepsilon_{to}$ , respectively; while those at the failure time,  $t_f$ , are shown by  $\varepsilon_f$  and  $\varepsilon_{tf}$ , respectively. The solution is dependent on the scaling exponent  $\alpha$ . For  $\alpha$ =1, the strain rate is

159

$$\varepsilon_t = \varepsilon_{to} e^{a(t-to)}. \tag{2}$$

In the introduction, the author notes that complexity may get in the way of precursor development. It is worth considering how much of this subsequent development is likely to apply to a complex (e.g., non-planar) fault in a complex (e.g., inhomogeneous) crustal setting. Detailed studies of earthquakes-possible in the past 20 years due to dense seismic and geodetic recording-show faults and earthquakes to be extremely complex at all scales studied.

162 For  $\alpha$ <1, the strain rate is 163  $\varepsilon_t = [a(1-\alpha)(t-t_o) + \varepsilon_{to}^{(1-\alpha)}]^{1/(1-\alpha)}.$ 164 (3) 165 166 For a>1, the strain rate is 167  $\varepsilon_t = [a(\alpha-1)](t_f-t) + \varepsilon_{tf}^{(1-\alpha)}]^{1/(1-\alpha)}.$ (4) 168 169 170 These equations remarkably reveal that  $\varepsilon_t$  increases with time and thus there is not an 171 upper bound of  $\varepsilon_t$ . The value of  $\varepsilon_t$  can be evaluated from the first two equations for 172  $\alpha \le 1$  and cannot be resolved from the third equation for  $\alpha > 1$ . It seems that there is a 173 singular point at  $t_f$  for  $\alpha > 1$ . At the singular point, a rock fracture or an earthquake 174 would happen. An example of numerical results can be seen in Voight's (1989) Figure 175 2. Since  $\varepsilon$  is integrated from  $\varepsilon_t$ , there is not an upper bound value for  $\varepsilon$  when  $\alpha \le 1$ . 176 We may further solve the time-dependent strain  $\varepsilon(t)$  through double integration of 177 Eq. (1). For  $\alpha > 1$  and  $\alpha \neq 2$ , the result is 178  $\varepsilon(t)-\varepsilon_o = \{ [a(\alpha-1)(t_f-t_o)+\varepsilon_{tf}^{(1-\alpha)}]^{\eta} - [a(\alpha-1)(t_f-t)+\varepsilon_{tf}^{(1-\alpha)}]^{\eta} \}/a(\alpha-2)$ 179 (5) 180 181 where  $\eta$  represents  $(2-\alpha)/(1-\alpha)$ . For  $\alpha > 1$  and  $\alpha \neq 2$ , the values of  $\eta$  are: (1)  $\eta < 0$  as  $1 < \alpha < 2$ ; and (2)  $\eta > 0$  as  $\alpha > 2$ . From the theoretical studies made by Main (1998), we 182 can see that the condition of the existence of accelerating strain for generating an 183 184 earthquake is  $1

of earthquake prediction proposed in this study is described below.

195

**3.1 Predicting the Failure Time of a Forthcoming Earthquake**

197

Since the condition  $1<\alpha<2$  is considered here, we will only take Eq. (4) in the followings. Due to  $1-\alpha<0$ , the strain rate,  $\varepsilon_{tf}$ , at the failure time should be much larger than 1 strain/sec and thus  $\varepsilon_{tf}^{1-\alpha}$  is much smaller than 1 strain/sec. This makes Eq. (4)

become

$$\varepsilon_t = [a(\alpha - 1)(t_f - t)]^{1/(1 - \alpha)}. \tag{6}$$

201202203

The time variations in  $\varepsilon_t$  from Eq. (6) for  $\alpha$ =1.5, 1.6, and 1.7 when  $\alpha$ =0.5 are displayed in Fig. 1 in which  $\varepsilon_t$  is normalized by the maximum value of  $\varepsilon_t$  for the three cases. In the figure, the three curves intersect to one another at a point with t= $t_c$ . When t<t,  $\varepsilon_t$  increases slowly with time and increases with  $\alpha$ ; while when t>t,  $\varepsilon_t$  increases rapidly with time and decreases with increasing  $\alpha$ .

From Eq. (6), we propose a method to explore the possibility of predicting the failure time,  $t_f$  of a forthcoming earthquake. Since the values of three model parameters  $t_f$ , a, and  $\alpha$ , must be solved, those of  $\varepsilon_t$  at three time instants should be given. Considering the pre-seismic strain rates, i.e.,  $\varepsilon_{t1}$ ,  $\varepsilon_{t2}$ , and  $\varepsilon_{t3}$ , at three time instants, i.e.,  $t_1$ ,  $t_2$ , and  $t_3$ , respectively. An example for  $\alpha$ =1.6 with  $\alpha$ =0.5 is shown in

Fig. 2 in which  $\varepsilon_t$  is normalized by the maximum value of  $\varepsilon_t$ . Inserting  $\varepsilon_{ij}$  and  $t_j$  (j=1,

2, and 3) into Eq. (6) yields

$$\varepsilon_{ij} = [a(\alpha - 1)(t_{j} - t_{j})]^{1/(1 - \alpha)} \quad (j = 1, 2, 3).$$
 (7)

$$t_f = t_j + \varepsilon_{ij}^{(1-\alpha)}/a(\alpha-1)$$
 (j=1, 2, 3). (8)

From Eq. (8) for  $\varepsilon_{t1}$  at  $t_1$  and  $\varepsilon_{t2}$  at  $t_2$ , we have

Much is made of 1

$$t_2$$
- $t_1$ - $t_2$ - $t_3$ - $t_3$ - $t_3$ - $t_4$ - $t_3$ - $t_4$ - $t_3$ - $t_4$ -

The difference between the occurrence time of a precursor and the failure time of the forthcoming earthquake is called the precursor time (e.g., Wang et al., 2016; Wang, 2021a,b) and is denoted by *T* hereafter. For the present case, the occurrence time of the precursor and the failure time of the forthcoming earthquake are  $t_o$  and  $t_f$ , respectively, thus leading to  $T=t_f-t_o$ .

**3.2 Prediction of the Magnitude of a Forthcoming Earthquake**

Based on the evaluated precursor time, T, it is possible to predict the magnitude of an earthquake by using Eq. (5). It first needs to discuss the value of initial strain  $\varepsilon_o$ . After the ruptures of last earthquake on a fault, the fault usually continues to slide with the relative movement speed of regional plates until the occurrence of the next event. If the moving speed is  $v_p$ , the strain rate,  $\varepsilon_t$ , is  $v_p/L$  where L is the fault length on a fault. Here the value of  $\varepsilon_t$  with the time unit  $\delta t$  of 1 second is taken to be  $\varepsilon_o$ . The value of  $\varepsilon_t$  is commonly  $10^{-6}$  strain/year around the world (e.g., Scholz et al., 1973; Turcotte and Schubert, 1982; Yu et al., 2001). For an example, the value of  $\varepsilon_t$  is  $0.25 \times 10^{-6}/\text{yr} = 1.90 \times 10^{-13}/\text{sec}$  for the San Andres fault (cf. Turcotte and Schubert, 1982), thus leading to  $\varepsilon_o = 1.90 \times 10^{-13}$  which is much smaller than 1. This makes us able to take  $\varepsilon_o = 0$  in this study. Figures 1 and 2 reveal  $\varepsilon_{tf} >> 1$ . According to the two conditions, Eq. (5) becomes

$$\varepsilon(t) = \{ [a(\alpha - 1)T]^{\eta} - [a(\alpha - 1)(t_f - t)]^{\eta} \} / a(\alpha - 2).$$
 (12)

Voight (1988, 1989) took a=0.5 for studying the results of his rock mechanic experiments. Hence, the values of  $\varepsilon_t$  is about a few strain/day or  $10^{-4}$  strain/sec for laboratory earthquakes. As mentioned above, the values of pre-seismic strain,  $\varepsilon$ , much before the occurrences of natural earthquakes are very small. Hence, the value of parameter a should be small for natural earthquakes. Nevertheless, the value of a is still taken to be 0.5 in Fig. 4 which illustrates the time variations in  $\varepsilon$  from Eq. (12) for  $\alpha$ =1.5, 1.6, and 1.7. In the figure,  $\varepsilon$  is normalized by the maximum value of the three cases. Like Fig. 1, Fig. 4 shows that the three curves intersect to one another at a point with t=t $\varepsilon$ . When t<tt $\varepsilon$ t increases slowly with time and increases with  $\alpha$ ; while

Why not use a realistic value of a, rather than a=0.5? This does not convince us that the treatment has relevance to the real world.

The earthquake ruptures at  $t=t_f$  when the strain is  $\varepsilon_f$ , which is when t>t,  $\varepsilon_t$  increases rapidly with time and decreases with increasing  $\alpha$ .

Here, as elsewhere, it would be helpful to provide an estimate of e-sub-f (the strain at initiation of failure) for reasonable values of a, alpha, nu and T. 289  $\varepsilon_f = [a(\alpha-1)T]^{\eta}/a(\alpha-2)$ (13)290 How do those estimates of e-sub-f compare to the 291 from Eq. (12). This is the upper bound of  $\varepsilon(t)$  for  $\alpha > 1$  and  $\alpha \neq 2$ . This upper bound is observational limits that we have on precursory strain? 292 dependent on both the parameters of fault-zone materials and precursor time. Many geodetic studies (including at Parkfield) have 293 Re-organizing Eq. (13) and taking the logarithm on the two sides of the re-organized shown that precursory strain 294 equation yield (if any) is below the detectable limit. 295 In other words, is the  $\log(T) = \log\{[a(\alpha-2)\varepsilon_f]^{1/\eta}/[a(\alpha-1)]^{\eta}\}.$ (14)296 author's analysis in line with 297 current observational data? If so, this is encouraging 298 Eq. (14) gives (though will emphasize that precursory strain should be 299 hard to detect). If not, the author should comment on  $\log(T) = \log\{[a(\alpha-2)]^{1/\eta}/a(\alpha-1)\} + \log(\varepsilon_f)/\eta.$ 300 (15)this point, and provide 301 arguments for why we should trust the following 302 This represents the power-law scaling relationship between T and  $\varepsilon_f$ , i.e.,  $T \sim \varepsilon_f^{1/\eta}$ . points in his analysis. 303 Since the rupture duration of an earthquake is short, we may consider  $\varepsilon_f$  as the 304 average strain over the ruptured area after failure. Shaw (2023) inferred the scaling law for  $\varepsilon_f$  versus the fault length of an earthquake, L, in the following form:  $\varepsilon_f = \lambda L^{-1/2}$ . 305 306 This leads to 307 308  $\log(\varepsilon_f) = \log(\lambda) - \log(L)/2$ , (16)309 Tell us the expected 310 where  $\lambda$  is a region-dependent constant. Several authors (e.g., Kanamori and range of values for this 311 Anderson, 1976; Wells and Coppersmith, 1994; Leonard, 2010; Thingbaijam et al., constant. Then calculate some examples for the 312 2017; Wang, 2018; Shaw, 2023) inferred the scaling law for L versus M, which is the expected values of esub-f, given this range of 313 earthquake magnitude (usually the seismic-wave magnitude,  $M_s$ , or the moment the contant and the 314 magnitude,  $M_w$ ), in the following form: length L of large-M earthquake ruptures 315 (e.g., hundreds of meters to tens of kilometers).  $\log(L) = \chi + M/2$ 316 (17)317 Again, tell us the 318 where  $\chi$  is a constant depending on tectonic and geological conditions. Combination expected range of this 319 of Eqs. (15), (16), and (17) leads to the log(T)–M relationship: constant. 320  $\log(T) = C + AM$ 321 (18)

where two new parameters are  $C = \log\{[a(\alpha-2)]^{1/\eta}/a(\alpha-1)\} + [\log(\lambda)-\chi/2]/\eta$  and  $A = -1/4\eta$ . Obviously, A is positive due to  $\eta

Hayakawa et al., 2006; Hayakawa and Hobara, 2010) suggest the goniometric method to assess the location of the forthcoming event by detecting the directions of ULF emissions from the observational stations to the earthquake epicenter. These methods seem acceptable.

Fig. 1 shows that the strain rate,  $\varepsilon_t$ , monotonically increases with time. From Fig. 1,

What does this mean? How is the author evaluating the "acceptability" of these hypotheses? Is he relying on his own intuition, or some theoretical arguments, or on published work that tested the hypotheses using real data?

**4 Discussion**

360361362

355356

**4.1 On the Theory for the Pre-seismic Strains**

364

369370

384

Eq. (1) will lead to an increase in the strain acceleration,  $\varepsilon_{tt}$ , with time. For the time variation as displayed in Fig. 1, at a certain time instant, larger  $\alpha$  yields higher  $\varepsilon_{\ell}$ . Meanwhile, there are two steps more or less separated at a particular time instant,  $t_c$ , which is shorter than  $t_f$  and not displayed in the figure. The two steps are:  $\varepsilon_t$  first slowly with time when  $t < t_c$  and then rapidly with time when  $t > t_c$ . Such a particular time appears earlier for large  $\alpha$  than for small  $\alpha$ . The second step is the existence of accelerating strain before a forthcoming earthquake from the theoretical studies by Main (1998). From observations of foreshocks, some authors (e.g., De Santis et al., 2015; and Cianchini et al., 2020) applied the revised accelerated moment release model to foreshocks revealing an acceleration pointing to the mainshock. Their model is similar to the present one. Since there is background noise in practical observations, the anomalous strain rate can be measured only in the second step. Like Fig. 1, Fig. 4 also illustrates the similar time variation in the strain,  $\varepsilon$ . For all cases in Fig. 4, there are also two steps separated at a particular time instant,  $t_c$ :  $\varepsilon$  first slowly with time when  $t < t_c$  and then rapidly with time when  $t > t_c$ . Unlike Fig. 1, such a particular time is almost the same for all  $\alpha$ 's in use. Meanwhile, in Fig. 4  $\varepsilon$  increases with  $\alpha$  when t is smaller than such a particular time; while  $\varepsilon$  decreases with increasing  $\alpha$ , when t is larger than such a particular time. This is the main difference between Fig. 1 and Fig. 4. In addition, larger  $\alpha$  produces lower  $\varepsilon_t$  as t is approaching  $t_f$  in Fig. 4. This means that the strain during a forthcoming earthquake increases with decreasing  $\alpha$ .

Missing word?

This statement is of questionable value without evaluating the level of "background noise" relative to expected amplitudes of anomalous strain during the two hypothesized steps. On what basis does the author conclude that the strain during the first step is below the "background noise" limit, and the strain during the second step is above that limit?

this study is basically similar to that used by Das and Scholz (1981) based the Charles

The theory of predicting the failure time of a forthcoming earthquake proposed by

are taken in this study, while only those of pre-slip at two time instants were considered in theirs. This is due to a reason that they assumed that the model parameters of either Charles law or Voight's equation have been already known, while those in this study are originally unknown and must be estimated from the observations.

Equation (18) exhibits the log(*T*)–*M* relationship based on pre-seismic strains.

law and that suggested by Main (1988) based on the Voight equation. One difference between this method and theirs is that the values of strain rate at three time instants

Using practical observations, how may the necessary parameters be estimated? Guidence would be helpful (perhaps provided in the following sections).

Equation (18) exhibits the  $\log(T)$ –M relationship based on pre-seismic strains. Tsubokawa (1969, 1973) first obtained a linear relation between the precursor time of crustal movement and mainshock magnitude for Japanese earthquakes in the form:  $\log(T)$ =-1.88+0.79M, with C=-1.88 and A=0.79. His observations somewhat confirm the existence of the  $\log(T)$ –M relationship. This makes us capable of predicting the magnitude of a forthcoming earthquake when the precursor time has been evaluated from observations. Although the earthquakes used by Tsubokawa (1969, 1973) occurred on different fault zones, his  $\log(T)$ –M relationship with the values of C and A represents the average characteristics of crustal deformations in Japan. In general, the parameters a and a0 of Voight's equation and a1 and a2 of the scaling laws of faults vary from area to area. Hence, the  $\log(T)$ –M1 relationships might be distinct in different fault systems.

If practical means exist to estimate the parameters, why does the author not do so using some examples of well-recorded earthquakes? Or can he point to any studies in which this was done in the 3-4 decades since those theoretical papers were published?

correct for Japan, compute the 403 precursory time T for various magnitudes of 404 Japanese earthquakes. Also, 405 use the analysis 406 above to estimate the amplitude of 407 precursory strain expected for various 408 earthquake 409 magnitudes. 410 Japan has the world's

Help the reader: what are the units of A and

C, and what units of T

Assuming that these values of A and C are are computed best geodetic by the current monitoring system.

Are those levels of geodetic network? If so, have they been detected, or not?

precursory strain large enough to be detected 413

(seconds?).

Gutenberg-Richter's energy-magnitude law of earthquakes (Gutenberg and Richter, 1942, 1956) is:  $\log(E_s)$ =11.8+1.5M in which  $E_s$  is the seismic-wave energy (in ergs) and M is commonly the surface-wave magnitude,  $M_s$ . From the law, he obtained the correlation:  $M \sim (2/3)\log(E_s)$ . In addition, from  $\log(T) = C + AM$  he got  $\log(T) \sim AM \sim (2A/3)\log(E_s)$ . Since  $E_s = \xi \Delta E$  where  $\Delta E$  is the strain energy of an earthquake and  $\xi$  (

**4.2.1 The log(T)-M relationships for other Precursors**

In order to measure the pre-seismic strains, the strain-meters should be installed on or much near the fault. When a strain-meter has not installed on or near the fault on which a forthcoming earthquake will happen, it is hence necessary to use other kinds of precursors which are directly or indirectly caused by the pre-seismic fault slip or strains for predicting the earthquake. In other word, it is much significant to explore the application of the present theory on the prediction of  $t_f$  and M of a forthcoming earthquake based on other kinds of precursors in practice. The present theory can be applied to other kinds of precursors, and thus the  $\log(T)-M$  relationships exist for these precursors. It is significant to apply the above-mentioned theory to predict the failure time and magnitude of a forthcoming earthquake based on other kinds of precursors.

The writing is generally quite clear, but the paper requires proofreading by a native english writer, to correct numerous small errors of grammar.

different kinds of precursors for a long time (Rikitake 1975a; Wang, 2021a,b, 2023; and cited references therein). From the plot of T (in days) versus M for five precursors, i.e., crustal movements, electric resistivity, radon (denoted as Rn hereafter) emission,  $v_p/v_s$  anomaly, and b-value of Gutenberg-Richter frequency-magnitude law

The  $\log(T)$ -M relationships have been recognized from the observations of

No verb in this sentence.

(Gutenberg and Richter, 1944). From 30 world-wide earthquakes, Scholz et al. (1973) Correct??? 439 inferred a relationship:  $M_s$ =-5.81+1.55log(T) (T in days) or log(T)=3.75+0.65 $M_s$ . For the precursors of crustal deformations and seismic-wave velocities, Whitcomb et al.

(1973) obtained  $\log(T) = 1.92 + 0.80M_s$  (*T* in days). Rikitake (1975b) obtained  $\log(T) = 1.92 + 0.80M_s$

$\frac{-1.83+0.76M_s}{(T \text{ in days})}$ . He also stressed that the  $\log(T)-M_s$  relationships are different for different groups of precursors. Rikitake (1979, 1984) divided a large data different for different groups of precursors. Rikitake (1979, 1984) divided a large data T in days or years?

set of 391 cases of precursors into three classes. He obtained log(T)=-1.01+0.60Ms for

No dependence on M?

the first class including 192 cases and log(T)=-1.0 for the second class. He did not report any relationship for the third class for foreshocks, tilt and strain, and earth's currents. Smith (1981, 1986) obtained the following relationship:  $log(T)=1.42+0.30M_s$  (T in years) from the data of abnormal b-values for earthquakes in New Zealand. Ding et al. (1985) obtained  $log(T)=-0.34+0.38M_s$  (T in years) for various precursors proceeding large Chinese earthquakes. From the b-value anomalies for 45 world-wide earthquakes with  $3 \le M_s \le 9$ , Wang et al. (2016) obtained

$\log(T) = (2.02 \pm 0.49) + (0.15 \pm 0.07) M_s (T \text{ in years}).$

From the previous description, it is clear that the log(T)-M relationships are

This paragraph summarizes findings from several papers. It should be straightforward for the author to provide a table of T for various values of M, for these studies.

For example, using the Scholz (1973) relation: for an earthquake of Ms=6, log(T)=3.75+0.65x6=7.65. T=4.47x10^7 days = 1,224 centuries. Is this a useful result?

For Whitcolm (1973) and Ms=6, T=759 days, which would be a more interesting hypothesis.

different for distinct kinds of precursors and also region-dependent. These results strongly suggest regional-dependence of C and A of Eq. (18). Clearly, C is influenced by several parameters, while A is controlled only by the scaling exponent,  $\alpha$ , of the fault-zone materials. Hence, A is an important indicator of the relationship. The previous studies lead to two interesting points. First, for the same forthcoming earthquake, different kinds of precursors may have different precursor times due to distinct values of C, but the same value of A. Secondly, for the forthcoming earthquakes that have the same magnitude and occur at different fault zones, different kinds of precursors may have different precursor times due to distinct values of both C and A.

It would be helpful to provde the reader with insight into the significance or meaning of A and C. That help is not provided when A and C are defined, nor here. And note that this discussion section located far later than Eqn 18. Without that insight, the reader won't appreciate the comments that follow, which seem important.

We will explore the theoretical basis for two kinds of precursors in the followings. The first kind of precursors is the geoelectric signals which are yielded almost within the fault zone where the forthcoming earthquake will happen, and the other is the geochemical signals which might occur on the sites that are somewhat far away from the fault zone. The mechanisms to generate the two kinds of signals will be described below.

**4.2.2** For the Geoelectric Precursors**

Changes or anomalies of geoelectric signals have been observed prior to earthquakes for a long time (cf. Hayakawa and Hobara, 2010; and cited references therein). Geoelectric signals are associated with pre-seismic slip on a fault where a forthcoming earthquake will happen. It is necessary to build up a comprehensive model that presents the lithosphere-ocean-atmosphere-ionosphere-magnetosphere coupling to interpret the generation of geoelectric precursors (Potirakis et al., 2017; Ouzounov et al., 2018; and cited references therein). Several proposed models are: (1) a model to present Rn ionization and charged aerosol and change of load resistance in the global electric circuit (Ouzounov et al., 2018; Pulinets and Ouzounov, 2018; and cited references therein); (2) a model to show coupling between stressed rocks and the atmosphere–ionosphere system (e.g., Kuo et al., 2011, 2014) based on experimental results of stress-induced charges made by Freund (2002); (3) a model to display ionosphere dynamics with imposed zonal (west-east) electric field (Zolotov et al., 2011, 2012; Namgaladze et al., 2012); and (4) a model of leakage of electric currents

Many studies have also failed to discover any EM anomalies prior to earthquakes.

[revised manuscript text omitted]

Provide some example calculations to give the reader a sense of the values of these various predictions.

In principle, the theory works well to predict the failure time of a forthcoming earthquake by using the pre-seismic geoelectric signals. But, in practice there might be a problem that the values of  $E_i$  cannot be observed accurately because of the presence of unexpected noise due to thunderstorm, atmospheric abnormal phenomena, and artificial effects. This problem should be very serious when  $t < t_c$  because their values are very small and cannot be observed. Hence, the observed data of geoelectric signals must be carefully selected and corrected to remove noise. The visible geoelectric signals should appear when  $t>t_c$  because the signals are strong enough. In

What is the basis for this statement? Has the author compared the amplitude of various noise sources to the expected signal amplitude of a precursor, and discovered that the precursor should be observable?

addition, in principle  $E_i$  must be measured near the fault. But, the monitoring station of geoelectric signals is usually not located near a fault where a forthcoming earthquake will happen. The value of  $E_i$  measured at a station not close to the near to expected epicenter should be slightly different from and weaker than near-fault one due to attenuation. Nevertheless, the attenuation of geoelectric signals measured at several time instants should be the same on the same station unless there are thunderstorm the levels of noise from and abnormal atmospheric phenomena between two time instants of different stations.

There have been several experiments in which EM or M networks have been established and maintained earthquake faults, or near an expected hypocenter. Papers based on those observations have explored various sources, and evaluated whether precursory signals are observed.

**4.2.3 For the Geochemical Precursors**

562

568

570

574

576

Numerous geochemical precursors are not observed at the localities near the earthquake epicenters (Wang 2021a,b; and cited references therein) because the observation stations are not installed at the sites near the epicenters. For example, Rn concentration anomalies prior to an earthquake are often observed somewhat far away from the epicenters because the measurement instruments are installed at hot-water springs or water-wells which may be far away from the epicenters. Nevertheless, their appearances are still related to the pre-seismic slip in the fault zones of forthcoming events. We assume that the presence of Rn concentration anomalies in the underground water might be associated with the spatial distribution of focal mechanism of an earthquake. The spatial pattern of the fault mechanism of an earthquake has four quarters: two for tension or dilatation and others for compression.

Kuo et al. (2010, 2019) reported a positive correlation between the temporal variation in Rn concentrations and that of dilatational strains measured at the Antong station for three events in southeastern Taiwan. The dilatational strains were related to tensional quarters of focal mechanisms of the events as mentioned above. They considered a model to explain Rn volatilization in an undrained fractured aquifer. This model is simply described below. A small fractured aquifer situated in a brittle rock, which is surrounded by a ductile formation in undrained conditions. When aquifer recharge is weak and negligible, undrained conditions are valid. There is only a single water phase in the aquifer before any precursory geochemical phenomenon appears. When the regional stress increases, dilation of brittle rock could occur at a faster rate reason to believe that than the rate of groundwater recharging into the newly created micro-cracks. As a result, gas saturation and two phases (gas and water) develop in the aquifer. The radon ahead of an earthquake, and in groundwater volatilizes into the gas phase and the Rn concentration in groundwater support that. If anything,

Or, because there are no anomalous signals to observe.

This is the key assumption that underlies so many of these precursor studies and theories. But there is little regional stress should increase and accelerate no observations that directly stress should decrease in the viscinity of any patch of accelerating precursory slip on the fault surface.

decreases. The model is mathematically represented by the following equation:

$$C_W/C_o = (HS_g + 1)^{-1}$$
 (21)

where  $C_o$  is the initial Rn concentration (in pCi/L) in formation brine (salt water);  $C_w$  is the equilibrium Rn concentration (in pCi/L) remaining in ground-water;  $S_g$  is the gas saturation (in %); H is Henry's coefficient (dimensionless) for Rn. From the rock-dilatancy model (Brace et al., 1966):  $\varepsilon_v = S_g/(1/\phi)$  or  $S_g = \varepsilon_v/\phi$  where  $\varepsilon_v$  and  $\phi$  denote, respectively, the (dimensionless) volumetric strain of the rocks beneath the observation site and the initial fracture porosity before rock dilatancy. The volumetric strain may be represented as  $\varepsilon_1 + \varepsilon_2 + \varepsilon_3$  where  $\varepsilon_j$  is the strain along the j-th axis (j=1, 2, and 3) (Turcotte and Schubert, 1982). This yields  $S_g = (\varepsilon_1 + \varepsilon_2 + \varepsilon_3)/\phi$ . Equation (19) shows that  $C_w$  increases with decreasing  $S_g$ . Inside the brittle rocks underneath the observation site,  $S_g$  increases with  $\varepsilon_v$ , thus leading to a decrease in  $C_w$ . The value of  $\varepsilon_v$  inside the brittle rocks underneath the observation site will be induced by the strain in the fault zone where the forthcoming earthquake will occur. Hence, the Rn concentration changes are controlled by pre-seismic strains that occur in the related fault zone.

Note that although we have considered a model to describe the production of preseismic geochemical signals, the production processes could be more complicated than the present model. Schirripa Spagnolo et al. (2024) addressed that preseismic geochemical signal are produced by the transport of chemical markers throughout the aquifers producing complex spatial circulations and alterations which can be extremely difficult to grasp using just one single model. They also claimed that such complex interactions among fault zones, host rocks upper and lower crustal volumes produce a wide range feedback mechanisms. These problems are beyond the scope of this study and need further investigations.

Of course, the time-dependent pre-seismic slip or strain on a fault along which a forthcoming earthquake will happen can produce stress changes surrounding the fault (Aki and Richards, 1980). This might induce some geochemical precursors which occur on some places somewhat far away from the fault. Hence, such kinds of precursors will appear more or less later than the pre-seismic slip or strain that happened on the fault. This results in a shorter precursor time than that for the

629

639

becomes pre-seismic slip or strain. Here, we consider a mechanical model to explain the problem. Dobrovolsky et al. (1979) used a half space, during the preparation processes of an earthquake, a zone of cracked rocks is formed in the focal area under the tectonic loading,  $\tau$ . The media inside the zone may be considered as a solid inclusion with different moduli that are lower than that of the half space. The solid inclusion re-distributes the stresses accompanied by deformations, including those on the Earth's ground surface. Let V be the solid soft inclusion volume that is an ellipse with a long-axis length of  $l_l$  and a short-axis length of  $l_s$ :  $l_l > 1$  for  $M \ge 5$  and  $l_l = l_s$  for M<5, thus leading to  $V=\pi l_1 l_s^2/6$  for  $M\geq 5$  and  $V=\pi l_s^3/6$  for M<5. The shear modulus of the half space and that of the inclusion are  $\mu$  and  $\mu$ - $\delta\mu$ , respectively. The ratio  $\delta\mu/\mu$  is denoted by  $\varphi$ . Assuming that the zone of effective manifestation of the precursory deformations is a sphere with the center at the epicenter of the forthcoming earthquake under the shear stresses loaded at infinity. In the spherical zone with a not specifically the radius of  $r_{\varepsilon}$ , the deformation has a strain being equal to or exceeding a certain  $\varepsilon_s$ which is smaller than the strain on the related fault. The  $r_{\varepsilon}$  is called the 'strain radius.' They obtained  $r_{\varepsilon}=0.85(\varphi V\tau/\mu\varepsilon_s)^{1/3}$ . This leads to

In most of the paper, the assumption is that there is precursory strain related to the entire future fault area. hypocenter or epicenter. The author should point this out, and explain why it is reasonable to consider these two contrasting views of precursory processes.

$$\varepsilon_s = (0.85)^3 \varphi V \tau / \mu r_{\varepsilon}^3. \tag{22}$$

This reveals that the strain decreases when the radius or the distance from the earthquake hypocenter increases. Based on Eq. (22), Rn concentration anomaly could occur at a distance  $r_{\varepsilon}$  from the hypocenter when the strain at the observation site is larger than  $\varepsilon_s$ . Hence, the pre-seismic strain in the related fault zone must be larger than a particular value,  $\varepsilon_p$  (> $\varepsilon_o$ ), at time  $t=t_p$ . This makes the occurrence time of Rn concentration anomaly be later than that of the pre-seismic strain because of  $t_p > t_o$ . Thus, the precursor time of the former is shorter than that of the latter. Equation (5)

If I follow correctly, this is a comparison between strain changes that scale with Ms, to Rn anomalies that are focussed around the hypocenter. Why is it not the case that e-sub-f is also concentrated at the future hypocenter?

[revised manuscript text omitted]

A lot of information has been presented in this paper. It would be helpful to add a short discussion section in which examples are given for how a scientist might practically proceed: Given an observational network (e.g., of strain, or Rn, or something else), what steps are required to obtain useful estimates of the future earthquake's time and magnitude?

Press. London, 477-525 1987.

Based on the theoretical results made by Main (1998) and the observed values of A of 720 the relationships, the value of  $\alpha$  must be in the range 1.0 to 1.8 for the generation of 721 earthquakes. The  $\log(T)$ -M relationships of pre-seismic geoelectromagnetic and 722 geochemical signals are taken into account. Theoretical results reveal that the 723 precursor times of the pre-seismic geoelectromagnetic precursors and those of 724 geochemical precursors are, respectively, the same and shorter than that of the 725 pre-seismic strains. 726 727 Data availability. No 728 729 Competing interests. There are no known competing financial interests or personal 730 relationships that could have appeared to influence the work reported in this paper. 731 732 Acknowledgments. This study was supported by the Institute of Earth Sciences, 733 Academia Sinica, Taiwan, ROC. 734 735 References 736 737 Aki, K.: Ideal probabilistic earthquake prediction, Tectonophys., 169, 197-198, DOI:10.1016/0040-1951(89)90193-5, 1989. 738 739 Aki, K.: Seismology of Earthquake and Volcano Prediction. Science Press Beijing, 740 China, 331 pp (with Chinese translation), 2009. 741 Aki, K. and Richard, P.G.: Quantitative Seismology. H. Freeman and Co., San 742 Francisco, 932pp, 1980. 743 Atkinson, R.K.: Subcritical crack growth in geological materials, J. Geophys. Res., 89, 744 4077-4114, 1984. 745 Atkinson, B.K.: Introduction to fracture mechanics and its geophysical applications, 746 In: Atkinson, B.K. (Ed.), Fracture Mechanics of Rock, 1-26, Academic Press, 747 London, 1987. 748 Atkinson, B.K. and Meredith, P.G.: Experimental fracture mechanics data for rocks and minerals, In: Atkinson, B.K. (Ed.), Fracture Mechanics of Rock, Academic 749

observations that may be used to estimate alpha for seismogenic crust?

Again, it's been 27 years since Main (1998). Has there been any work or

[revised manuscript text omitted]
_l(t)$ , and three values of  $\varepsilon_l(t)$ , i.e.,  $\varepsilon_{l2}$ ,  $\varepsilon_{l2}$ , and  $\varepsilon_{l3}$ , at three time instants,  $t_1$ ,  $t_3$ , and  $t_3$  for  $\alpha$ =1.6 when a=0.5.

 $\alpha^{-1.6}$

1077

Figure 3. The plot displays the curve for  $F_{2l}(\alpha)$ - $F_{3l}(\alpha)$ . The intersection point of the curve and the line with  $F_{2l}(\alpha)$ - $F_{3l}(\alpha)$ =0 is at  $\alpha$ =1.6.

10791080

Time

Figure 4. The plot shows the time variations in strain,  $\varepsilon(t)$ , for  $\alpha$ =1.5, 1.6, and 1.7 when  $\alpha$ =0.5. The three curves intersect one another at the point with t= $t_c$ .